# Bi-level Heterogeneous Learning for Time Series Foundation Models: A Federated Learning Approach

## Abstract

Heterogeneity in time series data is more pronounced than in vision or language, as temporal dynamics vary substantially across domains and tasks. Existing efforts on training time series foundation models (TSFMs) from scratch are often trained with mixed-batch strategies that merge large-scale datasets, which can cause gradient conflicts and degrade representation quality. To address this, we propose a fine-grained learning method that distills invariant knowledge from heterogeneous series while reducing cross-domain interference. We characterize heterogeneity at two levels: inter-domain and intra-domain. To tackle this bi-level heterogeneity, we design a federated learning method that mitigates intra-domain conflicts by enforcing domain-invariant and semantically consistent representations through local regularization, and addresses inter-domain discrepancies by enhancing cross-domain collaboration via domain-aware aggregation. Experiments across diverse benchmarks show that TSFMs trained with our method consistently outperform both centralized and federated TSFM baselines in point and probabilistic forecasting, while also achieving competitive zero-shot performance at scale, offering a flexible pathway for training TSFMs from scratch in heterogeneous environments.

## 1 Introduction

Time series forecasting plays a critical role in decision-making domains such as weather (Bi et al., 2023), energy (Wu et al., 2022), and urban computing (Zhang et al., 2024b). Recent advances in foundation models have shifted research from task-specific structures toward general-purpose forecasting models trained on diverse datasets (Goswami et al., 2024; Shi et al., 2024). However, existing time series foundation models (TSFMs) are often built under centralized training paradigms (Shi et al., 2024; Das et al., 2024; Woo et al., 2024), where heterogeneous datasets are simply merged for pretraining. In practice, this mixed-batch strategy can amplify gradient conflicts and obscure domain-specific structures, ultimately limiting the quality of learned representations. Moreover, temporal data are inherently fragmented across organizations and applications, making centralized collection impractical and further aggravating heterogeneity across diverse domains.

Concretely, these limitations manifest as a form of bi-level heterogeneity. The first level is inter-domain heterogeneity (Chen et al., 2025), where datasets from different sources exhibit covariate shifts and divergent temporal patterns due to variations in sensing modalities, sampling rates, and deployment environments. This misalignment often causes centralized or mixed-batch training to overfit domain-specific signals and fail to capture globally consistent dynamics. The second level is intra-domain conflict, which arises within a single dataset or domain (Wang et al., 2024b). Temporal concept drift from sensor aging, reconfiguration, or task evolution, together with latent sub-domains such as geographically distributed devices, leads to semantically similar but contextually divergent fragments. Naive training over such mixtures produces incoherent embeddings that are neither specialized nor aligned within sub-domains. The bi-level heterogeneity undermines the learning of domain-invariant and temporally coherent patterns, thereby limiting the generalization of TSFMs

A more flexible learning paradigm is required to handle bi-level heterogeneity in TSFMs training, one that can integrate knowledge across diverse domains while limiting cross-domain interference. Federated learning (FL) (McMahan et al., 2017) naturally offers such a strategy: by enabling de-

centralized training over distributed datasets, it allows domains to collaborate without centralizing raw data. Yet, existing FL methods primarily address inter-domain heterogeneity through personalization (Tan et al., 2022) or domain-aware adaptations (Yang et al., 2023), while typically assuming that each client's local data is homogeneous and stationary. While such assumptions may hold in static modalities like CV and NLP, they break down in time series domains where a single client can contain multiple latent sub-domains or undergo temporal concept drift. These intra-domain conflicts blur local representations, leaving them neither domain-specialized nor globally aligned. Recent work (Chen et al., 2025) explores task-agnostic FL for time series, but it largely targets inter-domain misalignment while overlooking intra-domain conflicts and evolving semantics. The open question is how to design a federated training paradigm that can simultaneously resolve bi-level heterogeneity, and in doing so, provide a scalable pathway for building robust TSFMs.

To this end, we propose FedTRL, a FL method for bi-level heterogeneous learning that enables domain-invariant time series representations under both inter-domain heterogeneity and intra-domain conflicts. Our FedTRL integrates two complementary components, including: (1) a domain-adversarial optimization objective combined with prototype alignment, which mitigates sub-domain drift and suppresses domain-specific artifacts to enforce semantically consistent local repre-

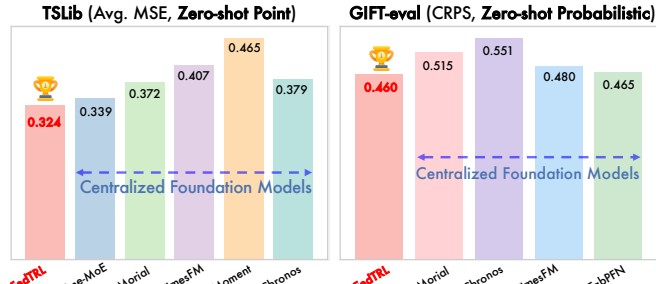

Figure 1: FedTRL outperforms centralized TSFMs in *zero-shot point* and *probabilistic forecasting*. Details are provided in Sec. 4

sentations, and (2) a domain-aware aggregation strategy that adaptively balances invariance and alignment across domains. FedTRL enables the training of robust TSFMs under decentralization. As shown in Fig. 1, FedTRL not only closes but surpasses centralized TSFMs in both zero-shot point (TSLib (Wu et al., 2022) as example) and probabilistic forecasting (GIFT-eval (Aksu et al., 2024) as example), demonstrating its effectiveness. Our main contributions are summarized as follows:

- We propose FedTRL, a FL method for bi-level heterogeneous learning that explicitly addresses both inter-domain and intra-domain heterogeneity challenges, providing a scalable and more flexible pathway for training time series foundation models from scratch.

- We proposed a fine-grained joint optimization–aggregation strategy that combines domain-adversarial regularization with prototype alignment and domain-aware aggregation, enforcing semantically consistent local representations while adaptively weighting client updates.

- Extensive experiments across representation learning, full/zero-shot forecasting, and probabilistic forecasting benchmarks show that FedTRL effectively scales FL to train TSFMs, achieving performance competitive with or even superior to centralized approaches.

## 2 RELATED WORK

**Unsupervised Time Series Representation Learning** Unsupervised representation learning aims to capture general temporal patterns from unlabeled sequences (Zhang et al., 2024a), providing the foundation for building TSFMs (*refer to Appendix A*), in contrast to task-specific supervised methods (Wu et al., 2022). Existing works mainly fall into masked reconstruction and contrastive learning. Masked reconstruction predicts missing inputs, with point-wise strategies such as TST (Zerveas et al., 2021), TimeMAE (Cheng et al., 2023), and SimMTM (Dong et al., 2023), and patch-wise variants like PatchTST (Nie et al., 2022) and CrossTimeNet (Cheng et al., 2025) that better capture local structures. Contrastive learning instead distinguishes positives from negatives, exemplified by TS-TCC (Eldele et al., 2021), which exploits temporal smoothness, and CoST (Woo et al., 2022), which leverages time–frequency augmentations. While effective on single datasets, these approaches face two key issues when scaled to TSFMs: *(i) dependence on dataset-specific fine-tuning* and *(ii) conflicting embeddings across heterogeneous domains*. This highlights the need for more flexible training paradigms that unify representation learning across domains, where FL (McMahan et al., 2017) offers a natural framework for collaborative pretraining on decentralized heterogeneous data.

**Federated Learning in Time Series.** Recent studies have extended FL to time series, exploring personalized cross-domain adaptation (Liu et al., 2024a; Soi et al., 2024) and task-agnostic modeling across domains (Chen et al., 2025; 2023; Sun et al., 2024). However, these methods often assume that local data within each client is homogeneous and domain-stable, an assumption rarely satisfied in practice. Most of them primarily target inter-domain heterogeneity across clients while overlooking intra-domain variability within individual clients. In reality, a single client may contain multiple latent sub-domains with divergent patterns, and cross-client distributions often remain misaligned despite similar marginal statistics. Such bi-level heterogeneity fundamentally disrupts representation learning, yielding incoherent local embeddings and misaligned global models that struggle to generalize across domains and tasks. To address these challenges, we propose FedTRL, a FL method for bi-level heterogeneous learning that explicitly tackles both inter-domain and intra-domain heterogeneity, enabling robust and scalable pretraining of unified TSFMs from scratch.

## 3 METHODOLOGY

**Problem Definition.** We consider an FL system with a server and $K$ clients (domains), where each client $k$ holds a local dataset $\mathcal{D}_k$ that may include data from one or multiple sub-domains. These datasets are heterogeneous due to differences in sampling rates, temporal dynamics, and domain-specific patterns. The objective is to collaboratively learn a unified representation that generalizes across domains and supports diverse forecasting tasks without supervision. The global objective is

$$\min_\theta F(\theta) := \sum_{k=1}^{K} \frac{n_k}{n} F_k(\theta_k; \mathcal{D}_k), \tag{1}$$

where $F_k$ is the local objective on client (domain) $k$, $n_k$ is its sample size, and $n = \sum_{k=1}^{K} n_k$. However, this standard objective fails to yield generalizable patterns because: (i) *inter-domain heterogeneity*, where distinct distributions across domains lead to divergent local optima $\theta_k^*$, making naive aggregation ineffective; and (ii) *intra-domain conflicts*, where a single domain may host multiple latent sub-domains, and contextual shifts create inconsistent or hybrid embeddings.

**Overview.** We propose FedTRL (as shown in **Fig. 2**), a FL method for bi-level heterogeneous learning that explicitly addresses both inter-domain and intra-domain heterogeneity. FedTRL integrates: **(i)** a fine-grained local optimization objective that enforces domain invariance through adversarial supervision and prototype alignment and **(ii)** an domain-aware aggregation mechanism that re-weights client updates by combining domain discrimination risk with representation alignment.

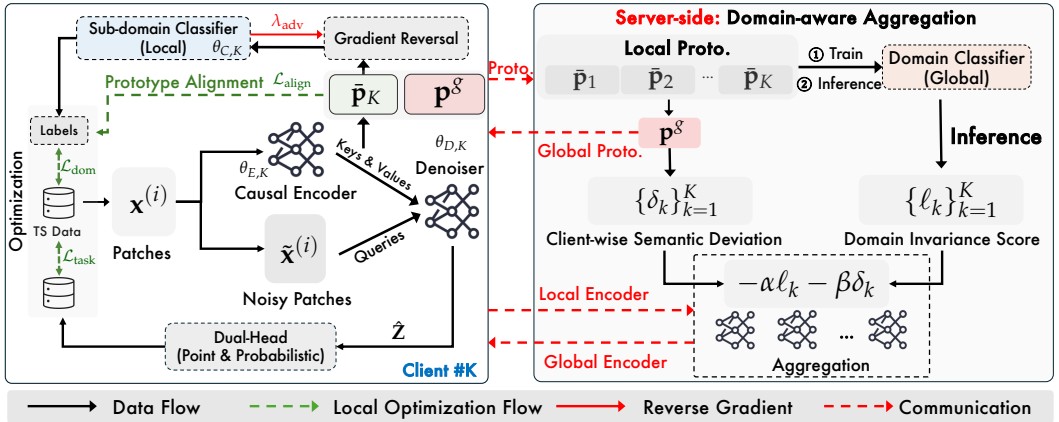

Figure 2: Structure of FedTRL (example of one client). Each client performs unsupervised diffusion-based reconstruction. In each communication, only Encoder $\theta_E$ and prototypes $\bar{\mathbf{p}}$ are uploaded to the server. The server performs Domain-aware aggregation (DaG) over both prototypes and encoder to obtain a unified encoder.

### 3.1 LOCAL UPDATING ON CLIENTS

To address heterogeneity at different levels, each client optimizes a unified objective: (i) reconstruction for temporal semantics and (ii) adversarial regularization for mitigating intra-domain conflicts,

while (iii) prototype alignment is designed to promote inter-domain consistency. Firstly, we decompose a multivariate time series $\mathbf{X} \in \mathbb{R}^{T \times C}$ of length $T$ into univariate series $\{\mathbf{x} \in \mathbb{R}^T\}$ along channels. Each univariate series is first stationarized using a non-parametric two-stage instance normalization (Liu et al., 2022) to mitigate temporal distribution shifts and anomalous ranges, thereby improving robustness across latent sub-domains. We then divide each sequence into non-overlapping patches of length $P$, yielding $N = \lfloor T/P \rfloor$ patches: $\mathcal{P} = \{\mathbf{x}^{(i)}\}_{i=1}^N$. This patching preserves local temporal structure while reducing computation for long sequences. Given these patches, we adopt diffusion reconstruction (Wang et al., 2025) to perform unsupervised representation learning. Specifically, we apply a forward diffusion process $\mathcal{D}_{\text{fwd}}$ to obtain its noisy counterpart:

$$\tilde{\mathbf{x}}^{(i)} = \sqrt{\bar{\alpha}_t}\, \mathbf{x}^{(i)} + \sqrt{1 - \bar{\alpha}_t}\, \boldsymbol{\epsilon}, \quad \boldsymbol{\epsilon} \sim \mathcal{N}(0, I),\ t \sim \mathcal{U}(0, T_d). \tag{2}$$

Unlike contrastive or masked reconstruction methods that implicitly assume data homogeneity, our approach prioritizes intrinsic temporal dynamics over domain-specific artifacts by jointly modeling point-, patch-, and sequence-level patterns. All clean patches are embedded and stacked into a token sequence: $\mathbf{E} \in \mathbb{R}^{N \times d}$, which is processed by a causal Transformer encoder $\theta_E$: $\mathbf{H} = \theta_E(\mathbf{E}) \in \mathbb{R}^{N \times d}$. In parallel, noisy patches $\tilde{\mathbf{x}}^{(i)}$ are embedded as query tokens $\tilde{\mathbf{E}} \in \mathbb{R}^{N \times d}$. A denoising decoder $\theta_D$ reconstructs latent signals using cross-attention mechanism, with queries $\tilde{\mathbf{E}}\mathbf{W}_q$ attending to encoder outputs $\mathbf{H}\mathbf{W}_k, \mathbf{H}\mathbf{W}_v$ as $\hat{\mathbf{Z}} = \theta_D(\tilde{\mathbf{E}}\mathbf{W}_q,\ \mathbf{H}\mathbf{W}_k,\ \mathbf{H}\mathbf{W}_v)$.

**Subdomain-Adversarial and Prototype-Aligned Regularization.** To mitigate conflicts among semantically related but contextually divergent sub-domains, we adapt subdomain-invariant learning via adversarial training (Ganin et al., 2016). We introduce a sub-domain classifier $D_{\theta_C}$ on clients, trained to predict the sequence-level sub-domain label $y^{\text{dom}}$, where each time series (and all patches derived from it) shares the same sub-domain category. Given encoder outputs $\mathbf{H}$, we compute the temporal prototype by mean pooling $\bar{\mathbf{p}} = \text{Mean}(\mathbf{H})$. The classifier objective is

$$\mathcal{L}_{\text{dom}} = \frac{1}{N} \sum\nolimits_{i=1}^N \text{CE}\big(D_{\theta_C}(\text{GRL}(\bar{\mathbf{p}}^{(i)})),\ y_i^{\text{dom}}\big), \tag{3}$$

where GRL (Gradient Reverse Layer) acts as identity forward and scales gradients by $-\lambda$ backward, forcing the encoder to suppress subdomain-specific signals. Here, $\lambda$ is a parameter that balances temporal modeling with subdomain invariance. Beyond local adversarial regularization, cross-domain knowledge transfer requires semantic alignment with a global reference. Instead of constraining full parameters, which are often entangled with domain-specific dynamics, we adopt a lightweight prototype-based alignment. Each client aligns its local prototype $\bar{\mathbf{p}}$ with the global prototype $\mathbf{p}^g$ as:

$$\mathcal{L}_{\text{align}} = ||\bar{\mathbf{p}} - \mathbf{p}^g||_2^2. \tag{4}$$

This ensures semantic consistency across clients while preserving local flexibility, yielding a domain-invariant representation space that facilitates robust aggregation.

**Point and Probabilistic Forecasting Support.** Each latent token $\hat{\mathbf{z}}^{(i)} \in \hat{\mathbf{Z}}$ is projected back to the patch space through an MLP, yielding reconstructed patches $\hat{\mathbf{x}}^{(i)}$. On top of these shared representations, we support both point and probabilistic forecasting within a unified dual-head architecture. The deterministic head outputs point predictions $\hat{y}_i$, optimized by mean square error (MSE), while the probabilistic head parameterizes a Student-t distribution with mean $\mu_i$, scale $\sigma_i$ and degrees of freedom $\nu$ (fixed $\nu = 5$ empirically) encouraging the model to capture heavy-tailed uncertainties via negative log-likelihood (NLL). Formally, the joint training objective can be formulated as

$$\mathcal{L}_{\text{task}} = \frac{1}{N} \sum_{i=1}^N (y_i - \hat{y}_i)^2 + \beta \cdot \frac{1}{N} \sum_{i=1}^N \left[ \frac{1}{2} \log(\pi(\nu - 2)\sigma_i^2) + \frac{\nu + 1}{2} \log\left(1 + \frac{(y_i - \mu_i)^2}{(\nu - 2)\sigma_i^2}\right) \right]. \tag{5}$$

where $y_i$ is the ground-truth value of the forecasting target derived from the original time series $\mathbf{x}$, and $\beta$ balances the deterministic and probabilistic objectives. To stabilize optimization under federated heterogeneity, we adopt a warm-up schedule: only the deterministic head is trained in the initial $R_{\text{warm}}$ rounds, after which the probabilistic head is gradually activated by annealing $\beta = 0$ to 1. This ensures accurate point forecasts first, followed by well-calibrated probabilistic predictions.

**Local Optimization Objective.** Each client jointly optimizes the three components introduced above. The reconstruction objective ensures temporal fidelity, the adversarial loss suppresses

subdomain-specific bias, and the prototype alignment term promotes semantic consistency across clients. The overall local training objective is therefore formulated as:

$$\mathcal{L} = \mathcal{L}_{\text{task}} + \lambda_{\text{dom}} \cdot \mathcal{L}_{\text{dom}} + \lambda_{\text{align}} \cdot \mathcal{L}_{\text{align}}, \tag{6}$$

where $\lambda_{\text{dom}}$ and $\lambda_{\text{align}}$ are balancing coefficients controlling the contribution of subdomain adversarial regularization and prototype alignment, respectively. Only the encoder $\theta_E$ and the local prototypes $\bar{\mathbf{p}}$ are shared with the server, as the goal is to collaboratively learn a domain-generalizable encoder. The decoder $\theta_D$ and sub-domain classifier $\theta_C$ are retained locally as auxiliary components to facilitate denoising and mitigate domain-specific bias. Their exclusion from aggregation prevents local semantics from interfering with the global encoder, enhancing generalization across domains.

## 3.2 Domain-aware Aggregation on Sever

Naive FL aggregation (McMahan et al., 2017) often dilutes discriminative signals and exacerbates domain entanglement under heterogeneous time series. To address this, we propose Domain-aware Aggregation (DaG), which integrates invariance and alignment into encoder aggregation. The server receives local encoders $\{\theta_{E,k}\}_{k=1}^{K}$ and prototypes $\{\bar{\mathbf{p}}_k\}_{k=1}^{K}$, and first updates a global prototype $\mathbf{p}_g$ via FedAvg. To measure invariance, we train a global domain classifier $f_d : \mathbb{R}^d \to \mathbb{R}^K$ using local prototypes, where the client (domain) index serves as the category label. After training, the classifier is fixed for inference. For each prototype $\bar{\mathbf{p}}_k$, the Domain Invariance Score is defined as $\ell_k = \mathcal{L}_{\text{CE}}(f_d(\bar{\mathbf{p}}_k), k)$: domain-specific prototypes yield low $\ell_k$, while domain-invariant ones approach uniform predictions and higher $\ell_k$. To complement this, we also measure semantic alignment with the global prototype (as Eq. 4), which captures residual domain drift beyond classifier predictions. We then integrate the two signals into a unified score: $s_k = -\alpha\ell_k - \beta\delta_k$, with $\alpha, \beta > 0$ balancing discriminability and alignment. Final aggregation is obtained via softmax weighting:

$$\theta_E^g = \sum_{k=1}^{K} \text{Softmax}(s_k) \cdot \theta_{E,k}. \tag{7}$$

By dynamically weighting client contributions based on both invariance and alignment, DaG prevents domain-dominant clients from skewing the global encoder and enhances cross-domain generalization. The global prototype $\mathbf{p}^g$ and updated global model $\theta_E^g$ are broadcast to clients for the next training round until convergence. The workflow description and algorithm are provided in **Alg. 1**.

## 4 Main Results

We evaluate FedTRL across multiple time series forecasting settings (Appendix B). First, we compare it with state-of-the-art unsupervised representation learning methods on in-domain forecasting (Section 4.1). We then assess its effectiveness in training large time series forecasting models (Section 4.2), as well as its full-shot adaption and zero-shot generalization (Section 4.2.1 and Section 4.2.2) on widely recognized benchmarks and real-world weather station datasets. Finally, we analyze FedTRL's benefits for unsupervised pretraining, conduct ablation and hyperparameter studies, and examine its scalability (Section 4.3). FedTRL proves that federated pretraining can scale into large time series foundation models, achieving competitive deterministic point or probabilistic forecasting performance while even surpassing advanced models with centralized training.

### 4.1 In-domain Point Forecasting

**Setup.** We conduct experiments on eight benchmark datasets from TSLib (Wu et al., 2022). Each dataset is treated as an independent client during federated training (without probabilistic head). For fine-tuning, we adapt the pretrained model to the corresponding target datasets. The look-back window is fixed at 512, and forecasting performance is evaluated across four horizons $\{96, 192, 336, 720\}$ using MSE and MAE. Model architectures and training configurations are summarized in Table 9.

**Results.** We compare FedTRL with three categories of baselines: (i) Masked reconstruction-based methods, including SimMTM (Dong et al., 2023) and PatchTST (Nie et al., 2022); (ii) Contrastive Learning-based methods, including TS-TCC (Eldele et al., 2021) and CoST (Woo et al., 2022); (iii) FL methods, including FFTS (Chen et al., 2025) and FedAvg (McMahan et al., 2017). Note that each client perform TimeDART(Wang et al., 2025) in FedAvg. Across eight benchmarks, FedTRL

achieves consistent improvements over both federated and centralized baselines. It not only surpasses existing federated pretraining approaches by clear margins, but also outperforms centralized training on several datasets, showing that properly aligned and aggregated FL can exceed centralized solutions. Its notable gains over FedAvg highlight strong robustness under heterogeneous clients.

Table 1: In-domain forecasting results (averaged across forecasting horizons $\{96, 192, 336, 720\}$). **Bold**: the best; Underline: the second best. The $^\dagger$ symbol denotes that based on the FedAvg aggregation protocol.

| Methods | OURS | | FEDERATED SELF-SUPERVISED$^\dagger$ | | | | | | | | FEDERATED FMs | | | | CENTRALIZED | |
| | FedTRL | | SimMTM | | PatchTST | | TimeMAE | | CoST | | FFTS | | FedAvg | | All Mixed | |
| Metric | MSE | MAE | MSE | MAE | MSE | MAE | MSE | MAE | MSE | MAE | MSE | MAE | MSE | MAE | MSE | MAE |
|---|---|---|---|---|---|---|---|---|---|---|---|---|---|---|---|---|
| ETTh1 | **0.448** | **0.472** | 0.495 | 0.512 | 0.478 | 0.495 | 0.502 | 0.518 | 0.521 | 0.540 | 0.463 | 0.485 | 0.476 | 0.495 | 0.458 | 0.488 |
| ETTh2 | **0.382** | **0.414** | 0.442 | 0.468 | 0.410 | 0.432 | 0.455 | 0.474 | 0.471 | 0.490 | 0.395 | 0.419 | 0.410 | 0.433 | 0.399 | 0.424 |
| ETTm1 | 0.375 | 0.402 | 0.415 | 0.439 | 0.386 | 0.408 | 0.421 | 0.444 | 0.436 | 0.455 | 0.380 | 0.406 | 0.399 | 0.406 | **0.372** | **0.398** |
| ETTm2 | **0.278** | **0.330** | 0.324 | 0.352 | 0.296 | 0.340 | 0.330 | 0.360 | 0.345 | 0.369 | 0.290 | 0.338 | 0.299 | 0.341 | 0.286 | 0.336 |
| Electricity | **0.181** | **0.272** | 0.216 | 0.306 | 0.192 | 0.288 | 0.225 | 0.315 | 0.242 | 0.329 | 0.187 | 0.280 | 0.198 | 0.293 | 0.189 | 0.285 |
| Traffic | 0.426 | 0.288 | 0.472 | 0.315 | 0.438 | 0.295 | 0.485 | 0.322 | 0.498 | 0.335 | **0.420** | **0.285** | 0.434 | 0.294 | 0.428 | 0.289 |
| Weather | **0.241** | **0.282** | 0.285 | 0.310 | 0.260 | 0.295 | 0.295 | 0.312 | 0.305 | 0.322 | 0.252 | 0.290 | 0.264 | 0.299 | 0.254 | 0.292 |
| Exchange | **0.375** | **0.423** | 0.460 | 0.480 | 0.405 | 0.445 | 0.468 | 0.490 | 0.472 | 0.492 | 0.390 | 0.440 | 0.403 | 0.444 | 0.390 | 0.438 |
| 1$^{st}$ Count | 12 | | 0 | | 0 | | 0 | | 0 | | 2 | | 0 | | 2 | |

## 4.2 SCALING TO TIME SERIES FOUNDATION MODELS

FedTRL can be scaled to learn a large TSFMs from cross-domain datasets, thereby enabling strong generalization capabilities on unseen data. To evaluate this capability, we pretrain a model on Time-MoE-300B (Shi et al., 2024), which spans nine domains and contains over 300 billion time points. Each domain is treated as an independent client, allowing FedTRL to learn cross-domain patterns in a federated setting. Note that all evaluation datasets were excluded from the pre-training dataset to ensure fairness. Model and training configurations for large-scale pretraining are shown in Table 9.

### 4.2.1 FULL/ZERO-SHOT POINT FORECASTING

We evaluate FedTRL on widely recognized forecasting benchmarks. Specifically, we adopt TS-FLib (Wu et al., 2022) for evaluating point forecasting. The ECL is excluded to avoid data leakage. Beyond standard benchmarks, we introduce RW-Bench, a collection of real-world weather datasets from 15 geographically diverse regions. Unlike curated academic corpora, RW-Bench retains noisy observations, providing a realistic testbed for zero-shot forecasting. Covering 10 variables at hourly resolution, it offers a timely (by Aug 2025) and challenging benchmark to evaluate TSFM's applicability under real-world conditions. Dataset statistics are provided in the Appendix B.4.

**Full-shot Adaption.** We benchmark FedTRL against advanced deep time series models, including TimeMixer (Wang et al., 2024a), TimeXer (Wang et al., 2024c), PatchTST (Nie et al., 2022), Times-Net (Wu et al., 2022), DLinear (Zeng et al., 2023), iTransformer (Liu et al., 2023), Autoformer (Wu et al., 2021), Non-Stationary Transformer (Liu et al., 2022), and LightTS (Zhang et al., 2022), as shown in Table 2. FedTRL-trained model consistently outperforms all baselines, showing notable advantages in both complex and simple domains despite requiring only one epoch of adaptation.

Table 2: Full-shot point forecasting results (averaged across $\{96, 192, 336, 720\}$). **Bold**: the best; Underline: the second best. Note that FedTRL-trained models are adapted to the target dataset with only one epoch.

| Models | FedTRL | | TimeMixer | | TimeXer | | iTransformer | | PatchTST | | TimesNet | | DLinear | | Autoformer | | Stationary | | LightTS | |
| Metrics | MSE | MAE | MSE | MAE | MSE | MAE | MSE | MAE | MSE | MAE | MSE | MAE | MSE | MAE | MSE | MAE | MSE | MAE | MSE | MAE |
|---|---|---|---|---|---|---|---|---|---|---|---|---|---|---|---|---|---|---|---|---|
| ETTh1 | **0.372** | **0.399** | 0.448 | 0.443 | 0.491 | 0.489 | 0.474 | 0.476 | 0.413 | 0.430 | 0.458 | 0.450 | 0.422 | 0.437 | 0.496 | 0.487 | 0.570 | 0.537 | 0.491 | 0.479 |
| ETTh2 | **0.330** | 0.383 | 0.364 | 0.394 | 0.356 | 0.398 | 0.372 | 0.411 | **0.330** | 0.379 | 0.414 | 0.427 | 0.431 | 0.446 | 0.450 | 0.459 | 0.526 | 0.516 | 0.602 | 0.543 |
| ETTm1 | **0.316** | **0.369** | 0.381 | 0.395 | 0.381 | 0.402 | 0.368 | 0.396 | 0.351 | 0.380 | 0.400 | 0.406 | 0.357 | 0.378 | 0.588 | 0.517 | 0.481 | 0.456 | 0.435 | 0.437 |
| ETTm2 | 0.257 | **0.313** | 0.275 | 0.323 | 0.275 | 0.329 | 0.271 | 0.331 | **0.255** | 0.315 | 0.291 | 0.333 | 0.267 | 0.333 | 0.327 | 0.371 | 0.306 | 0.347 | 0.409 | 0.436 |
| Electricity | **0.143** | **0.247** | 0.175 | 0.272 | 0.171 | 0.273 | 0.161 | 0.256 | 0.161 | 0.252 | 0.192 | 0.295 | 0.188 | 0.295 | 0.227 | 0.338 | 0.193 | 0.296 | 0.229 | 0.329 |
| Traffic | **0.355** | **0.256** | 0.405 | 0.284 | 0.419 | 0.302 | 0.385 | 0.275 | 0.390 | 0.263 | 0.620 | 0.336 | 0.433 | 0.295 | 0.628 | 0.379 | 0.624 | 0.340 | 0.622 | 0.392 |
| Weather | **0.214** | **0.252** | 0.241 | 0.272 | 0.227 | 0.268 | 0.235 | 0.274 | 0.255 | 0.264 | 0.259 | 0.287 | 0.248 | 0.300 | 0.338 | 0.382 | 0.288 | 0.314 | 0.261 | 0.312 |
| Exchange | **0.372** | **0.390** | 0.428 | 0.445 | 0.474 | 0.466 | 0.442 | 0.457 | 0.400 | 0.431 | 0.674 | 0.591 | 0.556 | 0.542 | 0.762 | 0.686 | 0.954 | 0.669 | 0.704 | 0.616 |
| 1$^{st}$ Count | 12 | | 0 | | 0 | | 0 | | 3 | | 0 | | 0 | | 0 | | 0 | | 0 | |

**Zero-shot Inference.** We evaluate the FedTRL-trained FM on TSLib and RW-Bench, following the evaluation protocol of (Shi et al., 2024). Results in Table 3 show that FedTRL consistently achieves state-of-the-art performance, delivering consistent MSE reductions across 20 datasets (5 in TSLib and 15 in RW-Bench). In particular, it demonstrates strong robustness on complex real-world weather station data, where temporal variability and distribution shifts are most pronounced. Notably, FedTRL-trained models outperform Time-MoE across all three reported scales, despite using fewer parameters than Time-MoE$_{ultra}$ (details about models are provided in Table 16). They also surpass TimesFM and Moment, which require additional fine-tuning for variable-length adaptation, whereas FedTRL operates fully zero-shot. This highlights that our optimization–aggregation design enables efficient pretraining of large time series foundation models under federated settings, producing representations that are both generalizable and well-calibrated across diverse domains. These results suggest that FedTRL not only matches but in some cases exceeds centralized paradigms, demonstrating the viability of federated pretraining as a powerful approach for TSFM training.

Table 3: Zero-shot point forecasting results (averaged across horizons $\{96, 192, 336, 720\}$ for observation lengths $\{512, 1024, 2048, 3072\}$). **Bold**: best; Underline: the second best. "–" means cannot be evaluated. The results on TSLib are officially from (Shi et al., 2024), the full results on RW-Bench are provided in Table 14.

| Models | FedTRL | | Time-MoE$_b$ | | Time-MoE$_l$ | | Time-MoE$_u$ | | Moirai$_s$ | | Moirai$_b$ | | Moirai$_l$ | | TimesFM | | Moment | | Chronos$_s$ | | Chronos$_b$ | | Chronos$_l$ | |
|---|---|---|---|---|---|---|---|---|---|---|---|---|---|---|---|---|---|---|---|---|---|---|---|---|
| Metrics | MSE | MAE | MSE | MAE | MSE | MAE | MSE | MAE | MSE | MAE | MSE | MAE | MSE | MAE | MSE | MAE | MSE | MAE | MSE | MAE | MSE | MAE | MSE | MAE |
| ETTh1 | 0.399 | 0.420 | 0.400 | 0.424 | **0.394** | **0.419** | 0.412 | 0.426 | 0.428 | 0.427 | 0.417 | **0.419** | 0.480 | 0.439 | 0.473 | 0.443 | 0.683 | 0.566 | 0.545 | 0.472 | 0.591 | 0.468 | 0.588 | 0.466 |
| ETTh2 | **0.350** | **0.380** | 0.366 | 0.404 | 0.405 | 0.415 | 0.371 | 0.399 | 0.361 | 0.384 | 0.362 | 0.382 | 0.367 | **0.377** | 0.392 | 0.406 | 0.361 | 0.409 | 0.424 | 0.430 | 0.405 | 0.410 | 0.455 | 0.427 |
| ETTm1 | **0.349** | **0.383** | 0.394 | 0.415 | 0.376 | 0.405 | 0.356 | 0.391 | 0.436 | 0.410 | 0.406 | 0.385 | 0.422 | 0.391 | 0.433 | 0.418 | 0.670 | 0.536 | 0.640 | 0.499 | 0.645 | 0.500 | 0.555 | 0.465 |
| ETTm2 | **0.282** | **0.330** | 0.317 | 0.365 | 0.316 | 0.361 | 0.288 | 0.344 | 0.307 | 0.347 | 0.311 | 0.337 | 0.329 | 0.343 | 0.328 | 0.346 | 0.316 | 0.365 | 0.349 | 0.380 | 0.310 | 0.350 | 0.295 | 0.338 |
| Weather | **0.238** | 0.279 | 0.265 | 0.297 | 0.270 | 0.300 | 0.270 | 0.300 | 0.275 | 0.286 | 0.287 | 0.281 | 0.264 | **0.273** | – | – | 0.294 | 0.326 | 0.300 | 0.318 | 0.292 | 0.315 | 0.279 | 0.306 |
| RW-Bench | **0.599** | **0.519** | 0.621 | 0.536 | 0.608 | 0.525 | – | – | 1.328 | 0.627 | 1.029 | 0.598 | 1.244 | 0.606 | 0.991 | 0.747 | 1.091 | 0.756 | 1.097 | 0.683 | 1.156 | 0.697 | 1.117 | 0.683 |
| $1^{st}$ Count | 8 | | 1 | | 2 | | 1 | | 0 | | 1 | | 2 | | 0 | | 1 | | 0 | | 1 | | 0 | |

### 4.2.2 ZERO-SHOT PROBABILISTIC FORECASTING

Beyond point forecasting, probabilistic forecasting is important to real-world applications with uncertainty. We experiment on GIFT-Eval (Aksu et al., 2024) and FEV leaderboard (Ansari et al., 2024), following their official evaluation suite and assessing point (MASE) and probabilistic (CPRS and WQL) metrics. All evaluated datasets are excluded from the pre-training dataset.

**GIFT-eval.** The evaluation results on GIFT-eval are shown in Table 4. This benchmark covers 23 datasets and 15 baselines, including statistical models, task-specific methods (DeepAR (Salinas et al., 2020), TiDE (Das et al., 2023a), N-BEATS (Oreshkin et al., 2020), PatchTST (Nie et al., 2022), iTransformer (Liu et al., 2023)), time series FMs (TimesFM (Das et al., 2024), TabPFN (Hoo et al., 2024), Chronos (Ansari et al., 2024)), and federated FMs (FedAvg (McMahan et al., 2017), FFTS (Chen et al., 2025)). Across all unseen datasets, FedTRL-trained model achieves the best performance in both MASE and CRPS. Remarkably, FedTRL also surpasses most advanced pretrained FMs, underscoring its effectiveness in large-scale FM pretraining for cross-domain time series.

Table 4: Zero-shot forecasting results on GIFT-eval, which comprises 32 datasets characterized by a variety of frequencies, variate numbers, and prediction lengths. We evaluate performance using 100 generated series, being consistent with (Woo et al., 2024). A lower MASE/CRPS indicates a better performance. Rank assigns a numerical ranking of all 97 configurations. Baseline results are officially reported by (Aksu et al., 2024).

| Type | Statistical Methods | | | | Supervised Task-Specific Models | | | | | Zero-Shot Models | | | | Zero-Shot Models (FL) | | |
|---|---|---|---|---|---|---|---|---|---|---|---|---|---|---|---|---|
| Model | Naïve | Seasonal Naïve | Auto ARIMA | Auto Theta | DeepAR | TiDE | N-BEATS | PTST. | iTrans. | TimesFM | TabPFN | Chronos | Moirai | FedAvg | FFTS | FedTRL (Ours) |
| **MASE** | 1.260 | 1.000 | 0.964 | 0.978 | 1.206 | 0.980 | 0.842 | 0.762 | 0.802 | 0.680 | 0.748 | 0.786 | 0.809 | 0.872 | 0.766 | **0.675** |
| **CRPS** | 1.383 | 1.000 | 0.770 | 1.051 | 0.721 | 0.652 | 0.689 | 0.496 | 0.524 | 0.465 | 0.480 | 0.551 | 0.515 | 0.700 | 0.521 | **0.460** |
| **Rank** | 28.072 | 26.175 | 21.515 | 24.031 | 18.938 | 18.557 | 21.381 | 10.052 | 11.320 | **8.237** | 8.268 | 14.309 | 10.175 | 18.231 | 9.787 | 8.676 |

**FEV Leaderboard.** We evaluate our FedTRL-trained model on the public FEV leaderboard established by AutoGluon (Ansari et al., 2024), which covers 27 datasets for probabilistic forecasting. As shown in **Fig. 3**, our model achieves zero-shot forecasting performance that surpasses most statistical and task-specific models trained with in-domain supervision. Among pretrained foundation

models, FedTRL ranks second overall (outperforming Chronos) and ranks first among federated FMs (outperforming FFTS and FedAvg). These improvements benefit from the diffusion-based reconstruction strategy, which integrates both patch- and point-wise reconstruction to capture fine-grained temporal dynamics. This enriched representation is further complemented by the dual-head design with a scheduling strategy that first secures accurate point forecasts before introducing uncertainty modeling. Together with the DaG-based aggregation that balances invariance and alignment across clients, these elements enable FedTRL to scale FL into an effective training recipe for large time series foundation models on deterministic point forecasting and probabilistic forecasting.

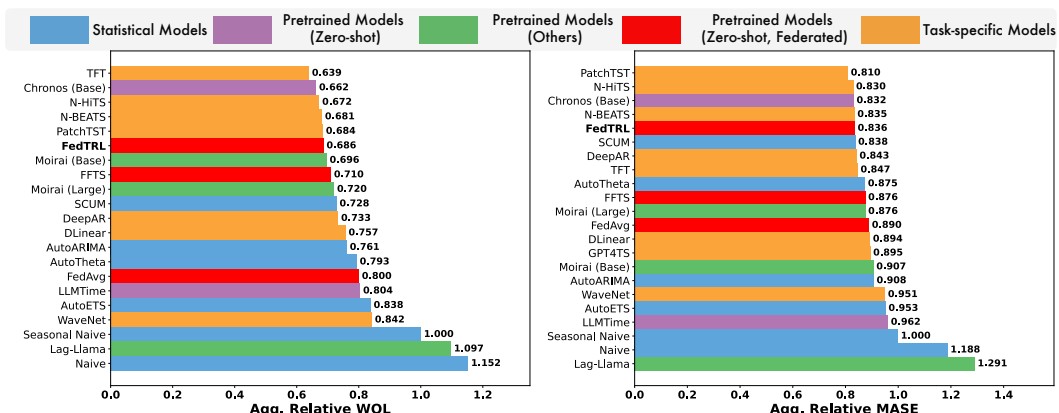

Figure 3: Results on FEV leaderboard. Baseline includes statistical methods, task-specific deep models trained on each dataset, and pre-trained foundation models. Pre-trained Models that have seen several datasets during pre-training are denoted as Pre-trained Models (Other). A lower MASE/WQL indicates a better result. FedTRL makes probabilistic predictions using 20 generated series, being consistent with (Ansari et al., 2024).

## 4.3 FRAMEWORK ANALYSIS

**Ablation Study.** We evaluate the contribution of three key components: (i) GRL-based adversarial domain regularization, (ii) prototype alignment, and (iii) domain-aware aggregation (DaG). Variants include *w/o* GRL, *w/o* Proto, and *w/o* DaG (replaced with vanilla FedAvg). Results on point forecasting is reported in **Table 5**, with probabilistic forecasting degradation shown in **Fig. 4**. Across both settings, removing any component consistently degrades performance. GRL stabilizes adversarial training and enforces subdomain invariance, yielding the largest gains in probabilistic metrics. Prototype alignment proves critical for bridging client-level representation gaps, especially in zero-shot transfer. DaG provides the strongest effect overall: replacing it with FedAvg leads to pronounced drops across all metrics, highlighting the necessity of domain-aware weighting in aggregation.

Table 5: Ablation study on point forecasting tasks.

| | Tasks | In-domain | | Full-shot | | Zero-shot | |
|---|---|---|---|---|---|---|---|
| | Variants | MSE | MAE | MSE | MAE | MSE | MAE |
| ETTh1 | **FedTRL** | **0.448** | **0.472** | **0.372** | **0.399** | **0.399** | **0.420** |
| | *w/o* GRL | 0.464 | 0.486 | 0.377 | 0.400 | 0.407 | 0.435 |
| | *w/o* Proto | 0.458 | 0.482 | 0.382 | 0.410 | 0.417 | 0.432 |
| | *w/o* DaG | 0.472 | 0.494 | 0.394 | 0.421 | 0.436 | 0.459 |
| ETTm1 | **FedTRL** | **0.375** | **0.402** | **0.316** | **0.369** | **0.350** | **0.380** |
| | *w/o* GRL | 0.388 | 0.414 | 0.324 | 0.372 | 0.359 | 0.384 |
| | *w/o* Proto | 0.382 | 0.408 | 0.320 | 0.372 | 0.361 | 0.384 |
| | *w/o* DaG | 0.400 | 0.420 | 0.333 | 0.364 | 0.380 | 0.399 |
| Weather | **FedTRL** | **0.241** | **0.282** | **0.214** | **0.252** | **0.238** | **0.279** |
| | *w/o* GRL | 0.260 | 0.295 | 0.217 | 0.257 | 0.240 | 0.287 |
| | *w/o* Proto | 0.252 | 0.290 | 0.222 | 0.259 | 0.251 | 0.282 |
| | *w/o* DaG | 0.254 | 0.292 | 0.239 | 0.275 | 0.289 | 0.292 |

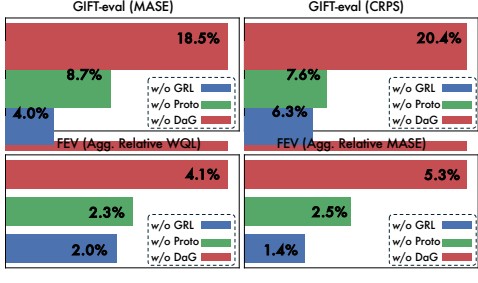

Figure 4: Relative performance drop of FedTRL ablations in zero-shot probabilistic forecasting, evaluated on GIFT-eval and FEV-leaderboard.

**Can FedTRL Improves Masked Reconstruction-based Methods?** To evaluate the generality of our proposed FedTRL, we integrate its key components into other masked reconstruction-based representation methods, including SimMTM and PatchTST. The results are reported in Table 6.

We observe that incorporating FedTRL consistently improves both MSE and MAE across four long-term forecasting datasets. These results demonstrate that our FedTRL can seamlessly enhance different reconstruction-based time series representation methods, highlighting its effectiveness and broad applicability.

Table 6: Results on ETT, Weather and Traffic datasets (averaging across four forecasting horizons). Improvement (%) is relative to the baseline.

| Model | ETTh1 | | ETTm1 | | Weather | | Traffic | |
|---|---|---|---|---|---|---|---|---|
| | MSE | MAE | MSE | MAE | MSE | MAE | MSE | MAE |
| SimMTM[†] | 0.495 | 0.512 | 0.415 | 0.439 | 0.285 | 0.310 | 0.472 | 0.315 |
| + FedTRL | 0.476 | 0.501 | 0.401 | 0.430 | 0.263 | 0.301 | 0.459 | 0.305 |
| **Improvement (%)** | **3.84** | **2.15** | **3.37** | **2.05** | **7.88** | **2.48** | **2.82** | **3.13** |
| PatchTST[†] | 0.478 | 0.495 | 0.386 | 0.408 | 0.260 | 0.295 | 0.438 | 0.315 |
| + FedTRL | 0.461 | 0.487 | 0.378 | 0.404 | 0.252 | 0.288 | 0.430 | 0.291 |
| **Improvement (%)** | **3.56** | **1.62** | **2.07** | **0.98** | **3.08** | **2.37** | **1.83** | **1.36** |

**Scalability.** Scalability is a key property for large time series foundation models. We evaluate the proposed FedTRL across different backbone sizes (from 38M to 302M, structure details in Table 16) to assess its scalability. Results (**Fig. 5**) consistently show that FedTRL maintains strong performance improvements on both point and probabilistic forecasting benchmarks as the backbone grows, demonstrating its robustness as a scalable training paradigm for models of varying capacity.

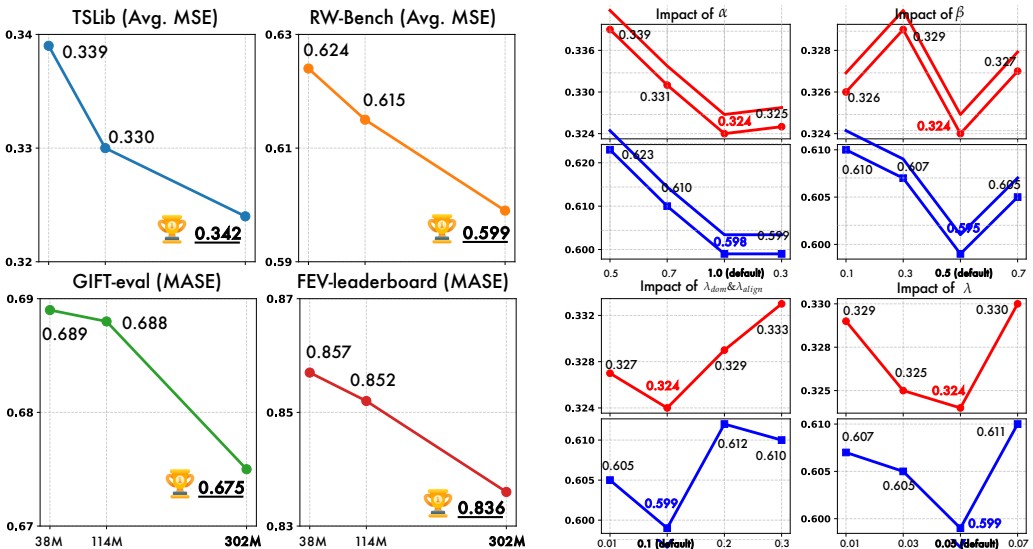

Figure 5: Model scalability across zero-shot both point forecasting and probabilistic forecasting benchmarks.

Figure 6: Hyperparameter sensitivity results (Avg. MSE). **Red Line**: TSLib; **Blue Line**: RW-Bench.

**Hyperparameter Sensitivity.** We examine hyperparameter sensitivity ($\alpha$, $\beta$, $\lambda_{\text{dom}}$&$\lambda_{\text{align}}$, $\lambda$) in **Fig. 6**, showing consistent trends across zero-shot point forecasting tasks. First, reducing $\alpha$ will gradually degrades performance, highlighting the importance of strong domain discriminability for aggregation. Second, $\beta$ exhibits a U-shaped trend: too small under-utilizes alignment, while too large over-regularizes, confirming that moderate values balance local and global consistency. Third, $\lambda_{\text{dom}}, \lambda_{\text{align}}$ are relatively stable within $[0.1, 0.3]$, but overly small values harm invariance learning. Finally, $\lambda$ also follows a concave trend, where weak reversal fails to suppress domain bias and strong reversal over-penalizes representations. These indicate that while FedTRL is robust within a range, its optimal defaults align well with the intended balance between discriminability and invariance.

## 5 CONCLUSION

We proposed FedTRL, a FL method for bi-level heterogeneous learning that tackles both inter- and intra-domain variability. By unifying domain-invariant local optimization with domain-aware aggregation, FedTRL learns patterns that remain consistent within sub-domains while generalizing effectively across domains. Experiments on multiple benchmarks covering point and probabilistic forecasting show that TSFMs trained with FedTRL not only outperform centralized and federated baselines, but also scale reliably to large configurations, achieving strong zero-shot generalization.

REPRODUCIBILITY STATEMENT

The used pretraining datasets are from TSLib (Wu et al., 2022), and Time-MoE-300B (Shi et al., 2024). Our experiments are conducted on widely used benchmarks, including TSLib (Wu et al., 2022), GIFT-eval (Aksu et al., 2024), and the FEV-leaderboard (Ansari et al., 2024), along with our proposed RW-Bench (details in Appendix B.4). Except for RW-Bench (will available in future), all datasets are publicly available. To facilitate reproducibility, we provide full implementation details, including training setups, model configurations and hyperparameter setups, in Appendix B.3 and Table 9. The workflow of FedTRL is outlined in Alg. 1, and we have release the complete code implementation at `https://anonymous.4open.science/r/FedTRL-ICLR-2026-Submission-E52A`.

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

## A  MORE RELATED WORK

**Time Series Foundation Models**  Pre-trained models that scale in capacity and data can evolve into foundation models capable of transferring across diverse domains. Moving beyond dataset-specific architectures, recent efforts have focused on building large time series foundation models (TSFMs). Through large-scale cross-domain pretraining, these models demonstrate strong zero-shot inference capabilities. For instance, ForecastFPN (Dooley et al., 2024) leverages synthetic data for zero-shot forecasting, CloudOps (Woo et al., 2023) applies masked modeling for domain-specific prediction, and TimeGPT-1 (Garza & Mergenthaler-Canseco, 2023) introduced the first commercial API. More recent advances such as MOMENT (Goswami et al., 2024), Moirai (Woo et al., 2024), TimeFM (Das et al., 2024), Chronos (Ansari et al., 2024), Timer (Liu et al., 2024b), and Time-MoE (Shi et al., 2024) scale pretraining to ultra-large datasets, achieving impressive cross-task transfer. However, most existing TSFMs are trained under centralized paradigms (Shi et al., 2024; Das et al., 2024; Woo et al., 2024), where heterogeneous datasets are simply merged into mixed batches. This practice often amplifies gradient conflicts and obscures domain-specific structures, leading to suboptimal representations. Moreover, temporal data are inherently fragmented across organizations and applications, making centralized collection impractical. These challenges call for more flexible training strategies that explicitly address bi-level heterogeneity, spanning both inter-domain misalignment and intra-domain variability. To this end, our work introduces a federated learning framework for bi-level heterogeneous learning, enabling scalable pretraining of TSFMs.

**Heterogeneous Federated Learning**  FL (McMahan et al., 2017) enables decentralized model training across distributed clients without sharing raw data. A key challenge is statistical heterogeneity, which degrades performance due to distributional shifts across clients. Early approaches reduce client–server divergence through alignment or regularization, but generally assume mild heterogeneity and shared feature spaces, an assumptions that break down in diverse real-world domains. To address this, Personalized Federated Learning (PFL) tailors models to individual clients, using strategies such as regularization-based decomposition(Hanzely et al., 2020; Li et al., 2021a), partial model sharing (Li et al., 2021b; Collins et al., 2021), adaptive aggregation (Zhang et al., 2020), or meta-learning (Fallah et al., 2020). While effective in vision and language tasks, these methods rely on transferable low-level features (Chen et al., 2025), which are less applicable to time series due to deeper variability in resolution, semantics, and physical context. More importantly, PFL prioritizes personalization over generalization, making it ill-suited for TSFM pretraining, where cross-domain transferability is crucial. This contrast is especially critical for time series: instead of producing personalized models, our goal is to train a unified TSFM that explicitly resolves bi-level heterogeneity, capturing both inter-domain discrepancies and intra-domain conflicts during pretraining.

## B  SUPPLEMENTARY MATERIALS

### B.1  GRADIENT REVERSAL LAYER

The Gradient Reversal Layer (GRL) (Ganin et al., 2016) is a standard technique in domain-adversarial learning. It acts as an identity function during the forward pass but reverses and scales gradients during backpropagation. Formally, given an input feature $\mathbf{h}$, GRL defines:

$$\mathrm{GRL}(\mathbf{h}) = \mathbf{h}, \quad \frac{\partial \mathrm{GRL}(\mathbf{h})}{\partial \mathbf{h}} = -\lambda I, \tag{8}$$

where $\lambda \geq 0$ is a tunable coefficient and $I$ denotes the identity matrix. This operation allows the sub-domain classifier to minimize its standard cross-entropy loss, while the encoder is simultaneously optimized to maximize the same objective, thereby learning domain-invariant representations. The coefficient $\lambda$ controls the strength of adversarial regularization, balancing domain invariance against the preservation of task-relevant temporal dynamics. Note that for training the local sub-domain classifier, we assign each sub-domain an index as its category label $y^{\mathrm{dom}}$, while for the global domain classifier, we use the client index as the category label.

### B.2  SUB-DOMAIN AND DOMAIN CLASSIFIER

The local sub-domain classifier promotes intra-domain invariance by mitigating fragmentation within each client, whereas the global domain classifier evaluates inter-domain separability to guide

adaptive aggregation. Together, they target both levels of heterogeneity and support robust, scalable pretraining of TSFMs. The specific details of these classifiers are described below.

**Local Sub-domain Classifier**    On each client, a sub-domain classifier $D_{\theta_C}$ is trained on $\bar{\mathbf{p}}$ with sub-domain index $y^{\text{dom}}$ as supervision. To encourage sub-domain–invariant representations, we apply a GRL to the encoder pathway and optimize the sub-domain classifier. So the classifier learns to predict sub-domains while the encoder learns to remove sub-domain cues (GRL scales gradients by $-\lambda$). This min–max game reduces the mutual information between embeddings and sub-domain labels, mitigating intra-domain drift without collapsing temporal content.

**Global Domain Classifier**    At the server level, we must evaluate how domain-specific each client's representation remains to guide aggregation. For this purpose, we train a global domain classifier $f_d : \mathbb{R}^d \rightarrow \mathbb{R}^K$, where $K$ is the number of clients (domains). Using client indices as supervision, $f_d$ is trained on client prototypes $\{\bar{\mathbf{p}}_k\}$. After training, $f_d$ is fixed and used for inference. For each prototype $\bar{\mathbf{p}}_k$, the Domain Invariance Score is defined as $\ell_k = \mathcal{L}_{\text{CE}}(f_d(\bar{\mathbf{p}}_k), k)$. Low $\ell_k$ indicates that the prototype is easily classifiable and thus domain-specific, while high $\ell_k$ suggests domain-invariance, as predictions approach uniform distribution. These scores, combined with semantic alignment to a global prototype, determine domain-aware aggregation weights.

### B.3    BASELINE AND BENCHMARK

**In-domain Point Forecasting.**    This task evaluates self-supervised time series representation learning. We introduce baselines across three categories: (i) masked reconstruction methods, including SimMTM (Dong et al., 2023) and PatchTST (Nie et al., 2022); (ii) contrastive learning methods, including TS-TCC (Eldele et al., 2021) and CoST (Woo et al., 2022); and (iii) federated learning methods, including FFTS (Chen et al., 2025) and FedAvg (McMahan et al., 2017), where each client adopts TimeDART (Wang et al., 2025). For fair comparison, all unsupervised baselines (except the FL ones) are implemented under the FedAvg protocol. In addition, a centralized baseline is constructed by mixing all datasets and training with the channel-independent (Nie et al., 2022). All methods adopt a consistent Transformer backbone. The results is shown in Table 1.

**Full-shot Point Forecasting.**    This task evaluates long-term point forecasting in a fully supervised setting, where models are trained or adapted using the entire training dataset. We benchmark FedTRL against advanced deep forecasting models, including TimeMixer (Wang et al., 2024a), TimeXer (Wang et al., 2024c), PatchTST (Nie et al., 2022), TimesNet (Wu et al., 2022), DLinear (Zeng et al., 2023), iTransformer (Liu et al., 2023), Autoformer (Wu et al., 2021), Non-Stationary Transformer (Liu et al., 2022), and LightTS (Zhang et al., 2022). For FedTRL, the model is first pretrained on Time-MoE-300B and then adapted to each target dataset with only a single epoch for fast transfer. The implementation is based on TSLib (Wu et al., 2022). Results are reported in Table 2.

**Zero-shot Point Forecasting.**    This task evaluates zero-shot long-term point forecastingg, where models are pretrained on large time series datasets. We benchmark FedTRL against advanced foundation models, including Time-MoE (Shi et al., 2024), TimesFM (Das et al., 2024), Morial (Woo et al., 2024), Chronos (Ansari et al., 2024), and Moment (Goswami et al., 2024). Evaluation is conducted on two real-world benchmarks: TSLib (four ETT-series, and Weather) and RW-Bench (15 datasets from diverse regions, details are provided in Section B.4). Following the protocol of (Shi et al., 2024), the look-back window is set to $\{512, 1024, 2048, 3072\}$ with corresponding prediction horizons $\{96, 192, 336, 720\}$. Among these baselines, TimesFM and Moment require fine-tuning for adaptation. The results on TSLib is officially from (Shi et al., 2024). Time-MoE$_{\text{ultra}}$ is excluded since it is not publicly available. The results is shown in Table 3 and full results in Table 13.

**Zero-shot Probabilistic Forecasting.**    This task evaluates zero-shot probabilistic forecasting. We evaluate FedTRL on two widely used benchmarks. GIFT-Eval (Aksu et al., 2024) comprehensively assesses performance across 23 datasets, covering 144K time series and 177M data points, with a total of 97 forecasting configurations. We follow the official evaluation protocol provided by Salesforce and report aggregated results in Table 4. In addition, we evaluate both accuracy and inference efficiency on the FEV leaderboard(Ansari et al., 2024), maintained by AutoGluon, which includes 27 datasets for standardized zero-shot testing. Aggregated results are reported in **Fig. 3**.

Table 7: Dataset statistics about TSLib (Wu et al., 2022). Channels indicates the number of time series (i.e., variables), and the size is organized in (training, validation, testing). [†]SE: Spectral Entropy; ACF: Autocorrelation-based; Proxy: Error-based. All three quantify time series forecastability, higher scores indicate greater predictability (i.e., lower task difficulty).

| Dataset | Domain | # Channels | Frequency | Size | Forecast Length | SE[†] | ACF[†] | Proxy[†] |
|---|---|---|---|---|---|---|---|---|
| ETTh1 | Power | 7 | Hourly | (8545, 2881, 2881) | $\{96, 192, 336, 720\}$ | 0.523 | 20.372 | 0.857 |
| ETTh2 | Power | 7 | Hourly | (8545, 2881, 2881) | $\{96, 192, 336, 720\}$ | 0.652 | 35.940 | 0.943 |
| ETTm1 | Power | 7 | 15 Minute | (34465, 11521, 11521) | $\{96, 192, 336, 720\}$ | 0.580 | 24.335 | 0.949 |
| ETTm2 | Power | 7 | 15 Minute | (34465, 11521, 11521) | $\{96, 192, 336, 720\}$ | 0.696 | 40.276 | 0.978 |
| Electricity | Energy | 321 | Hourly | (17805, 2537, 5166) | $\{96, 192, 336, 720\}$ | 0.706 | 12.137 | 0.814 |
| Traffic | Transportation | 862 | Hourly | (11673, 1661, 3413) | $\{96, 192, 336, 720\}$ | 0.567 | 6.977 | 0.681 |
| Weather | Weather | 21 | 10 Minute | (36792, 5271, 10540) | $\{96, 192, 336, 720\}$ | 0.549 | 25.818 | 0.842 |
| Exchange | Finance | 8 | 1 Day | (4704, 665, 1422) | $\{96, 192, 336, 720\}$ | 0.793 | 46.054 | 0.998 |

## B.4 RW-BENCH FOR ZERO-SHOT POINT FORECASTING EVALUATION

To assess the real-world applicability of pretrained FMs, we introduce Real-world Weather Benchmark (RW-Bench), a benchmark constructed from 15 ground-based weather observation stations collected in collaboration with our industry partners. Unlike curated academic datasets, RW-Bench preserves the complexity of raw operational data: no outliers are removed, and missing values are simply zero-filled. This design ensures that models are tested under conditions closer to real deployment scenarios. RW-Bench spans January 2022 to August 2025, providing timely and authentic observational records that complement the synthetic reanalysis data (e.g., ERA5) used in large-scale pretraining corpora such as Time-MoE-300B. *Importantly, the spatiotemporal non-overlap between RW-Bench and pretraining datasets enables a fair zero-shot evaluation of model generalization.* Each dataset contains hourly multivariate time series covering ten meteorological variables: air pressure, air temperature, relative humidity, precipitation, wind direction, wind speed, maximum wind direction, maximum wind speed, maximum temperature, and minimum temperature. Our RW-Bench present a challenging testbed for zero-shot forecasting. Deatiled statistic information and sample visualization from each stations are provided in Table 8 and Fig. 7, respectively.

Table 8: Dataset statistics about real-world weather benchmark (RW-Bench). Channels indicates the number of time series (i.e., variables), and the size is organized in (training, validation, testing). [†]SE: Spectral Entropy; ACF: Autocorrelation-based; Proxy: Error-based. All three quantify time series forecastability, higher scores indicate greater predictability (i.e., lower task difficulty).

| Dataset | Region | # Channels | Frequency | Size | Forecast Length | SE[†] | ACF[†] | Proxy[†] |
|---|---|---|---|---|---|---|---|---|
| RW-1 | Yuzhno-Sakhalinsk | 10 | Hourly | January 2022 - August 2025 | $\{96, 192, 336, 720\}$ | 0.438 | 17.089 | 0.682 |
| RW-2 | Tokyo | 10 | Hourly | January 2022 - August 2025 | $\{96, 192, 336, 720\}$ | 0.438 | 17.090 | 0.682 |
| RW-3 | Beijing | 10 | Hourly | January 2022 - August 2025 | $\{96, 192, 336, 720\}$ | 0.431 | 16.660 | 0.711 |
| RW-4 | Mumbai/Santacruz | 10 | Hourly | January 2022 - August 2025 | $\{96, 192, 336, 720\}$ | 0.398 | 16.474 | 0.706 |
| RW-5 | Ho Chi Minh City | 10 | Hourly | January 2022 - August 2025 | $\{96, 192, 336, 720\}$ | 0.462 | 17.292 | 0.768 |
| RW-6 | Cairo | 10 | Hourly | January 2022 - August 2025 | $\{96, 192, 336, 720\}$ | 0.456 | 16.912 | 0.734 |
| RW-7 | Nairobi Dagoretti | 10 | Hourly | January 2022 - August 2025 | $\{96, 192, 336, 720\}$ | 0.425 | 16.217 | 0.656 |
| RW-8 | Nauru | 10 | Hourly | January 2022 - August 2025 | $\{96, 192, 336, 720\}$ | 0.452 | 11.586 | 0.705 |
| RW-9 | Honolulu | 10 | Hourly | January 2022 - August 2025 | $\{96, 192, 336, 720\}$ | 0.435 | 11.696 | 0.734 |
| RW-10 | Los Angeles | 10 | Hourly | January 2022 - August 2025 | $\{96, 192, 336, 720\}$ | 0.425 | 15.879 | 0.726 |
| RW-11 | Edmonton | 10 | Hourly | January 2022 - August 2025 | $\{96, 192, 336, 720\}$ | 0.437 | 18.865 | 0.805 |
| RW-12 | Oxford | 10 | Hourly | January 2022 - August 2025 | $\{96, 192, 336, 720\}$ | 0.455 | 18.871 | 0.816 |
| RW-13 | Zurich | 10 | Hourly | January 2022 - August 2025 | $\{96, 192, 336, 720\}$ | 0.463 | 17.805 | 0.801 |
| RW-14 | McMurdo | 10 | Hourly | January 2022 - August 2025 | $\{96, 192, 336, 720\}$ | 0.431 | 16.438 | 0.720 |
| RW-15 | Novolazarevskaya | 10 | Hourly | January 2022 - August 2025 | $\{96, 192, 336, 720\}$ | 0.426 | 15.508 | 0.699 |

## B.5 MODEL ARCHITECTURE AND TRAINING CONFIGURATION

We adopt a vanilla Transformer (Vaswani et al., 2017) as the backbone. The local encoder is implemented as a causal Transformer encoder. For in-domain forecasting, we employ a standard multi-head attention mechanism with positional embeddings. For large-scale foundation model pretraining, we replace this with FlashAttention (Dao et al., 2022) and RoPE (Su et al., 2024) to improve efficiency, consistent with advanced foundation model practices. Details are provided in Table 9.

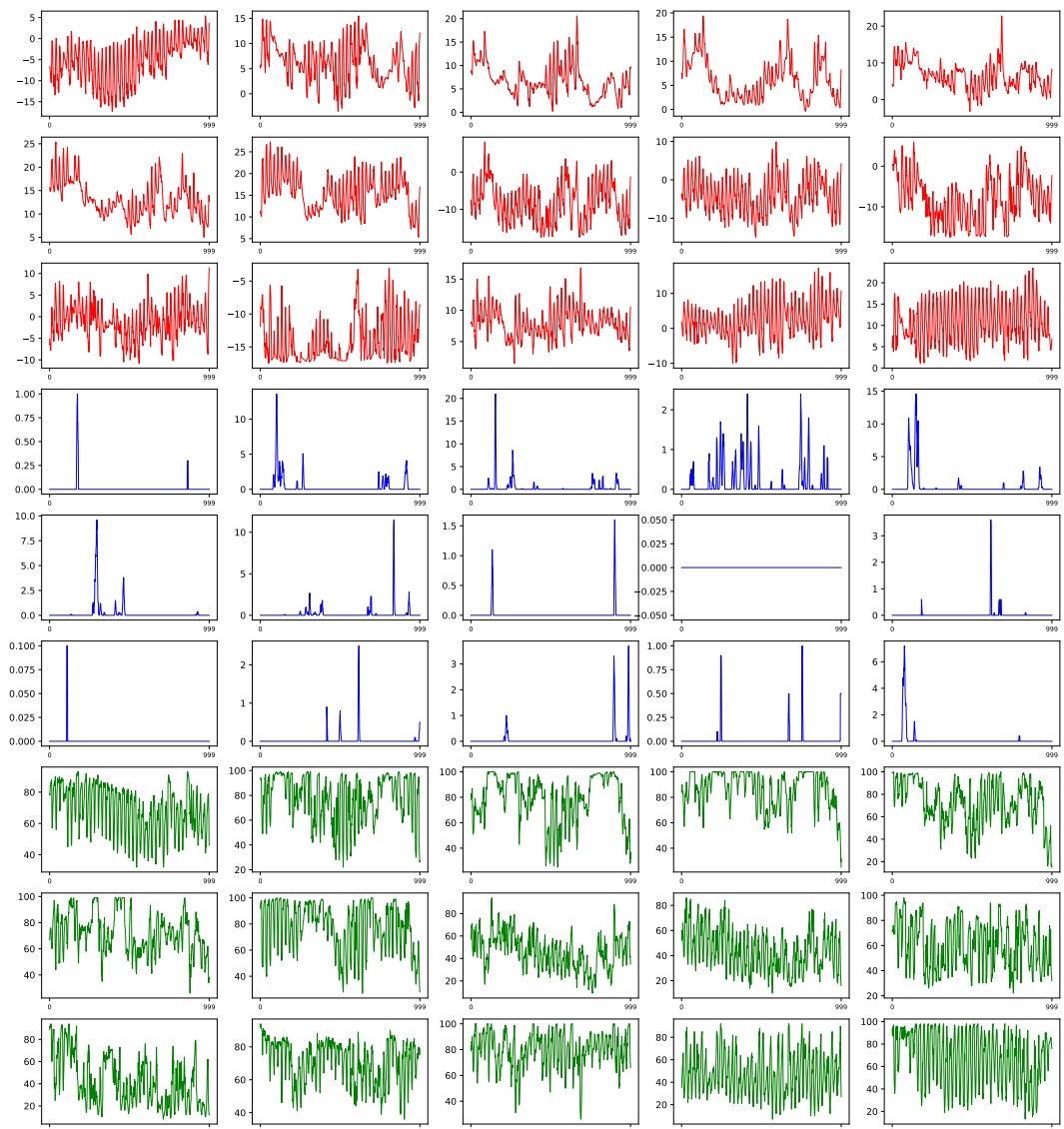

Figure 7: Visualization of RW-Bench samples: red denotes temperature, blue denotes precipitation, and green denotes humidity. Variables are arranged following the order in Table 8. RW-Bench will be open-sourced once the required permissions are obtained.

### B.6    WORKFLOW OF FEDTRL

The workflow of our proposed FedTRL as shown in Alg. 1. At each round, the server broadcasts the current global encoder and global prototype. Each selected client performs Diffusion-based reconstruction to learn temporal features, applies a local sub-domain classifier with a GRL to promote domain invariance, and aligns its prototype to the previous global prototype. The client then uploads only its encoder and prototype. The server forms a new global prototype via weighted averaging, trains a global domain classifier on client prototypes, and computes for each client a domain invariance score $(\ell_k)$ and a prototype deviation $(\delta_k)$ as the semnatic alignment vectors. These are combined into scores $s_k = -\alpha\ell_k - \beta\delta_k$, softmax-normalized to weights $s_k$, which are used to aggregate encoders into the updated global model. Decoder and sub-domain classifier remain local throughout, preventing client-specific bias from contaminating the global encoder.

Table 9: Detailed model architecture and training configuration.

| Category | Configuration (In-domain Forecasting) | Configuration (Large Foundation Model) |
|---|---|---|
| Optimizer | Adam | Adam |
| Batch Size | 128 | 2048 |
| Local Epochs | 10 | 100 |
| Global Rounds | 200 | 10,000 |
| Learning Rate (Local Updating) | 0.0001 | 0.0005 |
| Input Sequence Length | 512 | 3072 |
| Learning Scheduler | StepLR | StepLR |
| Diffusion Step | 1250 | 2500 |
| Computation Devices | $4 \times$ Nvidia RTX A5500-24GB GPUs | $16\times$ Nvidia A100-80G GPUs |
| Attention | Vanilia | FlashAttention with RoPE |
| Encoder Layer | 4 | 24 |
| (Denoising) Decoder Layers | 2 | 6 |
| Feedforward Dimension | 512 | 4096 |
| Model Dimension | 512 | 1024 |
| Number of Heads | 4 | 8 |
| Dropout | 0.4 | 0.4 |
| Activation Function | GELU | GELU |
| Patch Length | 16 | 16 |
| Stride | 16 | 16 |
| Domain Classifier Structure | $2 \times$ MLP with ReLU | $4 \times$ MLP with ReLU |
| Training Epoch on Server | 5 | 20 |
| Learning Rate | 0.0005 | 0.0001 |
| Domain Discriminability Weight $\alpha$ | 1.0 | 1.0 |
| Semantic Alignment Weight $\beta$ | 0.5 | 0.5 |
| Domain-adversarial Weight $\lambda_{\text{dom}}$ | 0.1 | 0.1 |
| Prototype Alignment Weight $\lambda_{\text{align}}$ | 0.1 | 0.1 |
| Warmup Coefficients $R_{\text{warm}}$ | Not applicable | 4,000 |

## C  FULL AND ADDITIONAL RESULTS

### C.1  MODEL SCALE DISCUSSION

Table 10 compares recent time series foundation models across architecture, scale, tokenization, and training setups. Centralized approaches such as Time-MoE, Moirai, and Chronos demonstrate strong zero-shot performance (**Tables 3 and 4, Figs. 3**), validating the effectiveness of large-scale pretraining on aggregated data. Unlike these centralized pipelines, FedTRL is pretrained in a fully federated manner, showing that models with hundreds of millions of parameters can be effectively trained without centralizing data. While comparable in scale to centralized FMs, FedTRL can yield flexible yet domain-invariant representations. Beyond matching performance, FedTRL exemplifies a paradigm shift in FM training, demonstrating that FL can serve not only as a privacy-preserving alternative but also as a viable strategy for building large-scale models across distributed domains.

### C.2  FEDERATED BASELINES UNDER HETEROGENEITY

A wide range of FL algorithms have been developed to address data heterogeneity (Tan et al., 2022), with strong results in CV and NLP. However, our experiments reveal that these methods transfer poorly to time series representation learning. Using the setup in Section 4.1, we evaluate representative approaches (FedProx (Li et al., 2020), FedPer (Arivazhagan et al., 2019), FedRep (Collins et al., 2021), and pFedMe (T Dinh et al., 2020)) on in-domain point forecasting, aggregating final-round local models for fair comparison. As shown in Table 11, only FedProx slightly improves over vanilla FedAvg, while others perform worse. We attribute this to fundamental differences between discrete classification benchmarks and continuous time series forecasting. Classification tasks have low-dimensional outputs where personalization strategies are effective, whereas forecasting requires modeling high-dimensional, continuous targets with complex temporal dependencies. Algorithms designed for discrete label spaces struggle to capture such dynamics, leading to poor generalization. In contrast, FedTRL directly addresses both inter- and intra-client heterogeneity through domain-invariant local objectives and domain-aware aggregation, yielding more robust TSFMs.

---

**Algorithm 1:** Workflow of FedTRL

---

**Input:** Initial encoder $\theta^{(0)}$; total rounds $T$; local epochs $E$; learning rate $\eta$; weighting factors $\lambda_{\text{domain}}, \lambda_{\text{align}}$

**Output:** Global encoder $\theta_E^{g,T}$ as the time series foundation model

**for** $t = 1, \ldots, T$ **do**

    Server selects a full client set $\mathcal{S}^t \subseteq \{1, \ldots, K\}$ of clients;

    **foreach** *client* $k \in \mathcal{S}^t$ *in parallel* **do**

        Download the last global encoder $\theta_E^{t-1}$ and prototype $\mathbf{p}^{t-1}$;

        **for** $e = 1$ **to** $E$ **do**

            Sample mini-batch $(\mathbf{X}_k, \mathbf{Y}_k) \sim \mathcal{D}_k$;

            Generate noised patches and perform denoising reconstruction;

            Compute reconstruction loss: $\mathcal{L}_{\text{task}}$;

            Compute domain loss via local classifier and GRL: $\mathcal{L}_{\text{dom}}$;

            Compute prototype deviation: $\mathcal{L}_{\text{align}}^t = \|\bar{\mathbf{p}}_k^t - \mathbf{p}^{g,t-1}\|^2$;

            Combine local optimization objective: $\mathcal{L}_k^t = (\mathcal{L}_{\text{task},k}^t + \lambda_{\text{dom}}\mathcal{L}_{\text{dom},k}^t + \lambda_{\text{align}}\mathcal{L}_{\text{align},k}^t)$;

            Update local encoder via: $\theta_{E,k}^t \leftarrow \theta_{E,k}^{t-1} - \eta \nabla_{\theta_{E,k}^t} \mathcal{L}_k^t$;

            (**in parallel**) Updated local decoder via: $\theta_{D,k}^t \leftarrow \theta_{D,k}^{t-1} - \eta \nabla_{\theta_{D,k}^t} \mathcal{L}_k^t$;

            (**in parallel**) Updated local classifier via: $\theta_{C,k}^t \leftarrow \theta_{C,k}^{t-1} - \eta \nabla_{\theta_{C,k}^t} \mathcal{L}_k^t$;

    Upload only $\theta_{E,k}^t$, $\bar{\mathbf{p}}_k^t$ (local prototype) to server for aggregation;

    // **Server-side aggregation**

    Aggregate prototypes: $\mathbf{p}^{g,t} =: \sum_k w_k \bar{\mathbf{p}}_k^t$;

    Updated and evaluate domain classifier on all $\mathbf{p}_k^t$, obtain domain invariance score $\ell_k^t$;

    Compute semantic alignment vectors: $\delta_k = \|\bar{\mathbf{p}}_k^t - \mathbf{p}^{g,t}\|^2$;

    Refine weights: $s_k = \text{Softmax}(-\alpha \cdot \ell_k^t - \beta \cdot \delta_k^t)$;

    Aggregate encoder: $\theta_E^t = \sum_k s_k \theta_{E,k}^t$;

    Broadcast new $\theta_E^{g,t}$, $\mathbf{p}^{g,t}$ to all clients;

**return** *Time series foundation model* $\theta_E^T$

---

Table 10: Comparison of time series foundation models. *Architecture* indicates the Transformer variant. *Model Size* reports parameter counts across scales. *Pre-training Scale* refers to the number of time points in pre-training datasets. *Token Level* specifies the granularity of time-series tokens. *Tokenization* describes which values are embedded from the series. *Context Length* denotes the maximum supported input length. *Probabilistic* indicates the ability to generate multiple possible predictions, in contrast to deterministic forecasters. *Training* refers to the training scenarios for this model, including centralized and federated (decentralized).

| Method | FedTRL (Ours) | Time-MoE (2024) | Moirai (2024) | MOMENT (2024) | LLMTime (2024) | Chronos (2024) | Lag-Llama (2023) | TimesFM (2023b) |
|---|---|---|---|---|---|---|---|---|
| Architecture | Encoder | Decoder | Encoder | Encoder | Decoder | EncDec | Decoder | Decoder |
| Model Size | 302M | 113M 453M 2.4B | 14M 91M 311M | 40M 125M 385M | – | 46M 200M 710M | 200M | 17M 70M 200M |
| Pre-training Scale | 300B | 300B | 231B | 1.13B | – | 84B | 0.36B | 100B |
| Token Level | Patch | Point | Patch | Patch | Point | Point | Point | Patch |
| Tokenization | Continuous | Continuous | Continuous | Continuous | Discrete | Discrete | Continuous | Continuous |
| Context Length | $\leq 3072$ | $\leq 4096$ | $\leq 5000$ | $= 512$ | - | $\leq 512$ | $\leq 1024$ | $\leq 512$ |
| Probabilistic | True | False | True | False | True | True | True | False |
| Training | **Federated** | Centralized | Centralized | Centralized | Centralized | Centralized | Centralized | Centralized |

## C.3 IN-DOMAIN FORECASTING

The full results of in-domain forecasting are shown in Table 12, where our proposed FedTRL consistently achieves the best performance across nearly all datasets and horizons. Compared to federated baselines, it shows clear error reductions and greater stability, while also rivaling or surpassing centralized training in several cases. These results highlight that FedTRL not only mitigates heterogeneity more effectively than existing FL methods but also delivers representations strong enough to match centralized pretraining, validating its robustness across diverse forecasting scenarios.

Table 11: In-domain forecasting results of FL baselines. **Bold**: the best.

| Dataset | FedTRL | FFTS | FedAvg | FedProx | FedPer | FedRep | pFedMe |
|---|---|---|---|---|---|---|---|
| ETT-h1 | **0.448** | 0.463 | 0.476 | 0.477 | 0.492 | 0.482 | 0.499 |
| ETT-m1 | **0.375** | 0.380 | 0.399 | 0.395 | 0.405 | 0.401 | 0.414 |
| Weather | **0.241** | 0.252 | 0.264 | 0.262 | 0.281 | 0.280 | 0.287 |

Table 12: Full results of in-domain forecasting (for four different forecasting horizons $\{96, 192, 336, 720\}$). **Bold**: the best; Underline: the second best. The $^\dagger$ symbol denotes that the federated unsupervised methods are based on the standard FedAvg (McMahan et al., 2017) aggregation protocol.

| Methods | | OURS FedTRL | | FEDERATED UNSUPERVISED$^\dagger$ SimMTM | | PatchTST | | TimeMAE | | CoST | | FEDERATED FMs FFTS | | FedAvg | | CENTRALIZED All Mixed | |
|---|---|---|---|---|---|---|---|---|---|---|---|---|---|---|---|---|---|
| Metric | | MSE | MAE | MSE | MAE | MSE | MAE | MSE | MAE | MSE | MAE | MSE | MAE | MSE | MAE | MSE | MAE |
| ETTh1 | 96 | **0.367** | **0.441** | 0.399 | 0.477 | 0.391 | 0.462 | 0.410 | 0.485 | 0.414 | 0.498 | 0.387 | 0.456 | 0.393 | 0.462 | 0.372 | 0.454 |
| | 192 | **0.433** | **0.464** | 0.466 | 0.503 | 0.449 | 0.482 | 0.476 | 0.507 | 0.500 | 0.531 | 0.444 | 0.481 | 0.454 | 0.490 | 0.437 | 0.478 |
| | 336 | **0.467** | **0.482** | 0.519 | 0.523 | 0.504 | 0.508 | 0.531 | 0.530 | 0.550 | 0.555 | 0.479 | 0.491 | 0.495 | 0.502 | 0.485 | 0.500 |
| | 720 | **0.525** | **0.501** | 0.595 | 0.546 | 0.569 | 0.528 | 0.591 | 0.549 | 0.620 | 0.576 | 0.542 | 0.511 | 0.562 | 0.527 | 0.539 | 0.520 |
| ETTh2 | 96 | **0.317** | **0.389** | 0.358 | 0.434 | 0.327 | 0.403 | 0.369 | 0.441 | 0.372 | 0.451 | 0.327 | 0.393 | 0.338 | 0.405 | 0.325 | 0.393 |
| | 192 | **0.371** | **0.411** | 0.416 | 0.460 | 0.397 | 0.427 | 0.439 | 0.468 | 0.443 | 0.481 | 0.383 | 0.417 | 0.400 | 0.433 | 0.381 | 0.419 |
| | 336 | **0.396** | **0.419** | 0.469 | 0.480 | 0.440 | 0.442 | 0.479 | 0.482 | 0.499 | 0.501 | 0.409 | 0.424 | 0.421 | 0.436 | 0.418 | 0.433 |
| | 720 | **0.445** | **0.437** | 0.526 | 0.497 | 0.476 | 0.456 | 0.532 | 0.505 | 0.570 | 0.526 | 0.462 | 0.442 | 0.481 | 0.458 | 0.472 | 0.452 |
| ETTm1 | 96 | 0.312 | 0.378 | 0.335 | 0.410 | 0.314 | 0.380 | 0.337 | 0.413 | 0.358 | 0.426 | 0.309 | 0.378 | 0.321 | 0.380 | **0.309** | **0.374** |
| | 192 | **0.351** | 0.395 | 0.396 | 0.433 | 0.363 | 0.402 | 0.405 | 0.439 | 0.408 | 0.447 | 0.369 | 0.403 | 0.385 | 0.404 | 0.352 | **0.392** |
| | 336 | 0.397 | 0.409 | 0.443 | 0.447 | 0.411 | 0.418 | 0.443 | 0.450 | 0.453 | 0.461 | 0.399 | 0.413 | 0.419 | 0.416 | **0.382** | **0.403** |
| | 720 | **0.440** | 0.426 | 0.486 | 0.466 | 0.456 | 0.432 | 0.499 | 0.474 | 0.525 | 0.487 | 0.442 | 0.430 | 0.471 | 0.424 | 0.445 | **0.423** |
| ETTm2 | 96 | 0.234 | 0.313 | 0.261 | 0.328 | 0.240 | 0.320 | 0.269 | 0.337 | 0.278 | 0.345 | 0.236 | 0.318 | 0.239 | 0.320 | **0.232** | **0.313** |
| | 192 | **0.269** | **0.325** | 0.313 | 0.347 | 0.285 | 0.336 | 0.310 | 0.353 | 0.330 | 0.363 | 0.280 | 0.333 | 0.287 | 0.337 | 0.272 | 0.331 |
| | 336 | **0.292** | **0.338** | 0.340 | 0.359 | 0.307 | 0.343 | 0.350 | 0.365 | 0.365 | 0.375 | 0.301 | 0.344 | 0.312 | 0.348 | 0.302 | 0.343 |
| | 720 | **0.317** | **0.344** | 0.382 | 0.374 | 0.353 | 0.361 | 0.390 | 0.384 | 0.407 | 0.393 | 0.343 | 0.356 | 0.360 | 0.359 | 0.339 | 0.356 |
| Electricity | 96 | **0.151** | **0.259** | 0.173 | 0.287 | 0.155 | 0.271 | 0.179 | 0.293 | 0.198 | 0.309 | 0.153 | 0.264 | 0.158 | 0.271 | 0.153 | 0.265 |
| | 192 | **0.173** | **0.269** | 0.209 | 0.302 | 0.185 | 0.282 | 0.213 | 0.309 | 0.229 | 0.323 | 0.178 | 0.275 | 0.188 | 0.283 | 0.182 | 0.283 |
| | 336 | **0.189** | **0.274** | 0.230 | 0.313 | 0.202 | 0.295 | 0.240 | 0.323 | 0.255 | 0.336 | 0.195 | 0.283 | 0.209 | 0.296 | 0.200 | 0.291 |
| | 720 | **0.211** | **0.287** | 0.251 | 0.322 | 0.225 | 0.304 | 0.268 | 0.335 | 0.286 | 0.349 | 0.221 | 0.298 | 0.237 | 0.322 | 0.221 | 0.301 |
| Traffic | 96 | 0.355 | 0.272 | 0.384 | 0.293 | 0.354 | 0.275 | 0.395 | 0.300 | 0.394 | 0.310 | **0.339** | **0.265** | 0.343 | 0.269 | 0.352 | 0.272 |
| | 192 | 0.408 | 0.283 | 0.453 | 0.310 | 0.422 | 0.291 | 0.462 | 0.316 | 0.478 | 0.331 | **0.395** | **0.281** | 0.405 | 0.288 | 0.414 | 0.285 |
| | 336 | 0.451 | 0.295 | 0.505 | 0.324 | 0.461 | 0.300 | 0.511 | 0.329 | 0.533 | 0.346 | **0.444** | **0.294** | 0.460 | 0.304 | 0.452 | 0.296 |
| | 720 | **0.491** | 0.303 | 0.547 | 0.334 | 0.516 | 0.314 | 0.572 | 0.343 | 0.588 | 0.353 | 0.502 | **0.301** | 0.528 | 0.316 | 0.495 | 0.304 |
| Weather | 96 | **0.201** | **0.266** | 0.229 | 0.288 | 0.208 | 0.277 | 0.237 | 0.289 | 0.248 | 0.303 | 0.204 | 0.271 | 0.228 | 0.285 | 0.210 | 0.277 |
| | 192 | **0.231** | **0.279** | 0.276 | 0.308 | 0.252 | 0.293 | 0.280 | 0.310 | 0.290 | 0.318 | 0.244 | 0.288 | 0.252 | 0.295 | 0.239 | 0.286 |
| | 336 | **0.254** | **0.286** | 0.302 | 0.319 | 0.277 | 0.301 | 0.310 | 0.317 | 0.316 | 0.326 | 0.262 | 0.296 | 0.270 | 0.304 | 0.266 | 0.297 |
| | 720 | **0.277** | **0.297** | 0.333 | 0.325 | 0.303 | 0.309 | 0.353 | 0.332 | 0.365 | 0.342 | 0.298 | 0.305 | 0.306 | 0.312 | 0.301 | 0.308 |
| Exchange | 96 | **0.311** | **0.397** | 0.376 | 0.449 | 0.326 | 0.414 | 0.374 | 0.452 | 0.385 | 0.459 | 0.319 | 0.410 | 0.332 | 0.414 | 0.322 | 0.410 |
| | 192 | **0.358** | **0.419** | 0.442 | 0.474 | 0.388 | 0.439 | 0.449 | 0.485 | 0.446 | 0.483 | 0.367 | 0.432 | 0.380 | 0.436 | 0.367 | 0.430 |
| | 336 | **0.393** | **0.429** | 0.477 | 0.485 | 0.431 | 0.458 | 0.493 | 0.499 | 0.502 | 0.503 | 0.409 | 0.446 | 0.430 | 0.452 | 0.413 | 0.448 |
| | 720 | **0.438** | **0.446** | 0.545 | 0.512 | 0.475 | 0.470 | 0.556 | 0.523 | 0.555 | 0.523 | 0.465 | 0.472 | 0.470 | 0.476 | 0.458 | 0.464 |
| 1$^{st}$ Count | | **49** | | 0 | | 0 | | 0 | | 0 | | 7 | | 0 | | 8 | |

## C.4 ZERO-SHOT POINT FORECASTING

We evaluate our FedTRL-trained model on two zero-shot point forecasting benchmarks: TSLib (5 datasets) and RW-Bench (15 datasets), following the evaluation protocol of (Shi et al., 2024). The full results are reported in Table 13 (TSLib) and Table 14 (RW-Bench). Across both benchmarks and multiple prediction horizons, FedTRL consistently achieves state-of-the-art performance, surpassing centralized foundation models trained on aggregated data. These further demonstrate

FedTRL's effectiveness. In addition, we provide comparisons with recent TSFM models, including VisionTS (Chen et al., 2024) and Sundial (94B/230B/1032B) (Liu et al., 2025), using the same zero-shot evaluation protocol as Table 3. The results are shown in Table 15 below, which indicates that our FedTRL-trained TSFM continues to outperform these models under the same evaluation setting.

Table 13: Full results of zero-shot forecasting experiments. A lower MSE or MAE indicates a better prediction. TimesFM, due to its use of Weather datasets in pretraining, is not evaluated on this dataset and is denoted by a dash (−). **Bold**: the best, Underline: the second best.

| Models | | Ours FedTRL | | \multicolumn{22}{c}{Pretrained Time Series Foundation Models (Zero-shot)} | | | | | | | | | | | | | | | | | | | |
| | | | | Time-MoE$_{base}$ | | Time-MoE$_{large}$ | | Time-MoE$_{ultra}$ | | Moirai$_{small}$ | | Moirai$_{base}$ | | Moirai$_{large}$ | | TimesFM | | Moment | | Chronos$_{small}$ | | Chronos$_{base}$ | | Chronos$_{large}$ | |
| | Metrics | MSE | MAE | MSE | MAE | MSE | MAE | MSE | MAE | MSE | MAE | MSE | MAE | MSE | MAE | MSE | MAE | MSE | MAE | MSE | MAE | MSE | MAE | MSE | MAE |
|---|---|---|---|---|---|---|---|---|---|---|---|---|---|---|---|---|---|---|---|---|---|---|---|---|---|
| ETTh1 96 | | **0.346** | 0.381 | 0.357 | 0.381 | 0.350 | 0.382 | 0.349 | **0.379** | 0.401 | 0.402 | 0.376 | 0.392 | 0.349 | **0.379** | 0.414 | 0.404 | 0.688 | 0.557 | 0.466 | 0.409 | 0.440 | 0.393 | 0.441 | 0.390 |
| 192 | | 0.386 | 0.407 | **0.384** | **0.404** | 0.388 | 0.412 | 0.395 | 0.413 | 0.388 | 0.412 | 0.412 | 0.413 | 0.434 | 0.415 | 0.465 | 0.434 | 0.688 | 0.560 | 0.530 | 0.450 | 0.492 | 0.426 | 0.502 | 0.424 |
| 336 | | 0.420 | 0.443 | **0.411** | 0.434 | **0.411** | 0.430 | 0.447 | 0.453 | 0.433 | 0.428 | 0.433 | 0.428 | 0.495 | 0.445 | 0.503 | 0.456 | 0.675 | 0.563 | 0.570 | 0.486 | 0.550 | 0.462 | 0.576 | 0.467 |
| 720 | | 0.444 | 0.449 | 0.449 | 0.477 | **0.427** | 0.455 | 0.457 | 0.462 | 0.439 | 0.454 | 0.447 | **0.444** | 0.611 | 0.510 | 0.511 | 0.481 | 0.683 | 0.585 | 0.615 | 0.543 | 0.882 | 0.591 | 0.835 | 0.583 |
| Avg. | | 0.399 | 0.420 | 0.400 | 0.424 | **0.394** | **0.419** | 0.412 | 0.426 | 0.428 | 0.427 | 0.417 | 0.419 | 0.480 | 0.439 | 0.473 | 0.443 | 0.683 | 0.566 | 0.545 | 0.472 | 0.591 | 0.468 | 0.588 | 0.466 |
| ETTh2 96 | | 0.300 | 0.352 | 0.305 | 0.359 | 0.302 | 0.354 | **0.292** | 0.352 | 0.297 | 0.336 | 0.294 | **0.330** | 0.296 | **0.330** | 0.315 | 0.349 | 0.342 | 0.396 | 0.307 | 0.356 | 0.308 | 0.343 | 0.320 | 0.345 |
| 192 | | **0.335** | **0.369** | 0.351 | 0.386 | 0.364 | 0.385 | 0.347 | 0.379 | 0.368 | 0.381 | 0.365 | 0.375 | 0.361 | 0.371 | 0.388 | 0.395 | 0.354 | 0.402 | 0.376 | 0.401 | 0.384 | 0.392 | 0.406 | 0.399 |
| 336 | | **0.371** | **0.392** | 0.391 | 0.418 | 0.417 | 0.425 | 0.406 | 0.419 | 0.370 | 0.393 | 0.376 | 0.390 | 0.390 | 0.390 | 0.422 | 0.427 | 0.356 | 0.407 | 0.408 | 0.431 | 0.429 | 0.430 | 0.492 | 0.453 |
| 720 | | **0.394** | **0.407** | 0.419 | 0.454 | 0.537 | 0.496 | 0.439 | 0.447 | 0.411 | 0.426 | 0.416 | 0.433 | 0.423 | 0.418 | 0.443 | 0.454 | 0.395 | 0.434 | 0.604 | 0.533 | 0.501 | 0.477 | 0.603 | 0.511 |
| Avg. | | **0.350** | **0.380** | 0.366 | 0.404 | 0.405 | 0.415 | 0.371 | 0.399 | 0.361 | 0.384 | 0.363 | 0.382 | 0.367 | 0.377 | 0.392 | 0.406 | 0.361 | 0.409 | 0.424 | 0.430 | 0.405 | 0.410 | 0.455 | 0.427 |
| ETTm1 96 | | 0.284 | 0.342 | 0.338 | 0.368 | 0.309 | 0.557 | **0.281** | **0.341** | 0.418 | 0.392 | 0.363 | 0.356 | 0.380 | 0.361 | 0.361 | 0.370 | 0.654 | 0.527 | 0.511 | 0.423 | 0.454 | 0.408 | 0.457 | 0.403 |
| 192 | | 0.319 | 0.366 | 0.353 | 0.388 | 0.346 | 0.381 | **0.305** | **0.358** | 0.431 | 0.405 | 0.388 | 0.375 | 0.412 | 0.383 | 0.414 | 0.405 | 0.662 | 0.532 | 0.618 | 0.485 | 0.567 | 0.477 | 0.530 | 0.450 |
| 336 | | **0.358** | 0.405 | 0.381 | 0.413 | 0.373 | 0.408 | 0.369 | **0.395** | 0.433 | 0.412 | 0.416 | 0.400 | 0.434 | 0.400 | 0.445 | 0.429 | 0.672 | 0.537 | 0.683 | 0.524 | 0.662 | 0.525 | 0.577 | 0.481 |
| 720 | | **0.435** | **0.419** | 0.504 | 0.493 | 0.475 | 0.477 | 0.469 | 0.472 | 0.462 | 0.432 | 0.460 | **0.418** | 0.462 | 0.420 | 0.512 | 0.471 | 0.692 | 0.551 | 0.748 | 0.566 | 0.900 | 0.591 | 0.660 | 0.526 |
| Avg. | | **0.349** | **0.383** | 0.394 | 0.415 | 0.376 | 0.405 | 0.356 | 0.391 | 0.436 | 0.410 | 0.406 | **0.385** | 0.422 | 0.391 | 0.433 | 0.418 | 0.670 | 0.536 | 0.640 | 0.499 | 0.645 | 0.500 | 0.555 | 0.465 |
| ETTm2 96 | | 0.205 | 0.279 | 0.201 | 0.291 | **0.197** | 0.286 | 0.198 | 0.288 | 0.214 | 0.288 | 0.211 | 0.274 | 0.202 | **0.270** | 0.211 | 0.274 | 0.260 | 0.335 | 0.209 | 0.291 | 0.199 | 0.274 | **0.197** | 0.271 |
| 192 | | 0.245 | 0.321 | 0.258 | 0.334 | 0.250 | 0.322 | **0.235** | **0.312** | 0.284 | 0.332 | 0.275 | 0.316 | 0.281 | 0.318 | 0.289 | 0.321 | 0.289 | 0.350 | 0.280 | 0.341 | 0.261 | 0.322 | 0.254 | 0.314 |
| 336 | | 0.311 | **0.343** | 0.324 | 0.373 | 0.337 | 0.375 | **0.293** | 0.348 | 0.331 | 0.362 | 0.329 | 0.350 | 0.341 | 0.355 | 0.360 | 0.366 | 0.324 | 0.369 | 0.354 | 0.390 | 0.326 | 0.366 | 0.313 | 0.353 |
| 720 | | **0.367** | **0.377** | 0.488 | 0.464 | 0.480 | 0.461 | 0.427 | 0.423 | 0.402 | 0.408 | 0.437 | 0.411 | 0.485 | 0.428 | 0.462 | 0.430 | 0.394 | 0.409 | 0.553 | 0.499 | 0.455 | 0.439 | 0.416 | 0.415 |
| Avg. | | **0.282** | **0.330** | 0.317 | 0.365 | 0.316 | 0.361 | 0.288 | 0.344 | 0.307 | 0.347 | 0.311 | 0.337 | 0.329 | 0.343 | 0.328 | 0.346 | 0.316 | 0.365 | 0.349 | 0.380 | 0.310 | 0.350 | 0.295 | 0.338 |
| Weather 96 | | **0.149** | **0.210** | 0.160 | 0.214 | 0.159 | 0.213 | 0.157 | 0.211 | 0.198 | 0.222 | 0.220 | 0.217 | 0.199 | 0.211 | – | – | 0.243 | 0.255 | 0.211 | 0.243 | 0.203 | 0.238 | 0.194 | 0.235 |
| 192 | | **0.188** | **0.240** | 0.210 | 0.260 | 0.215 | 0.266 | 0.208 | 0.256 | 0.247 | 0.265 | 0.271 | 0.259 | 0.246 | 0.251 | – | – | 0.278 | 0.329 | 0.263 | 0.294 | 0.256 | 0.290 | 0.249 | 0.285 |
| 336 | | **0.249** | **0.282** | 0.274 | 0.309 | 0.291 | 0.322 | 0.255 | 0.290 | 0.283 | 0.303 | 0.286 | 0.297 | 0.274 | 0.291 | – | – | 0.306 | 0.346 | 0.321 | 0.339 | 0.314 | 0.336 | 0.302 | 0.327 |
| 720 | | 0.366 | 0.384 | 0.418 | 0.405 | 0.415 | 0.400 | 0.405 | 0.397 | 0.373 | 0.354 | 0.371 | 0.351 | **0.337** | **0.340** | – | – | 0.350 | 0.374 | 0.404 | 0.397 | 0.397 | 0.396 | 0.372 | 0.378 |
| Avg. | | **0.238** | 0.279 | 0.265 | 0.297 | 0.270 | 0.300 | 0.256 | 0.288 | 0.275 | 0.286 | 0.287 | 0.281 | 0.264 | **0.273** | – | – | 0.294 | 0.326 | 0.300 | 0.318 | 0.292 | 0.315 | 0.279 | 0.306 |
| 1ˢᵗ Count | | 22 | | 3 | | 5 | | 9 | | 1 | | 7 | | 7 | | 1 | | 1 | | 0 | | 0 | | 1 | |

## C.5 ZERO-SHOT PROBABILISTIC FORECASTING

We evaluate our FedTRL-trained model on GIFT-Eval (Aksu et al., 2024), a benchmark designed to comprehensively assess forecasting across diverse time series. It contains 23 datasets with 144,000 series and 177 million data points, covering 97 forecasting configurations. Following the official evaluation suite, we report aggregated results in Table 4. We further assess zero-shot forecasting and inference efficiency on FEV leaderboard (Ansari et al., 2024), established by AutoGluon, which includes 27 datasets. Aggregated forecasting metrics are presented in Fig. 3.

## C.6 SCALABILITY

We evaluate the scalability of FedTRL by training models of different sizes, with configurations summarized in Table 16 and results reported in Fig. 5. We further assess scalability with respect to dataset size by controlling the amount of data used for pretraining across {90B, 120B, 180B, 300B, 450B, 540B}, where the 450B and 540B configurations are augmented using ERA5 daily, weekly, and monthly data. We evaluate the resulting TSFMs on full-shot, zero-shot, and zero-shot probabilistic forecasting tasks. As shown in Table 17, performance consistently improves with larger pretraining datasets, demonstrating the positive scaling behavior of FedTRL.

## C.7 ADDITIONAL DISCUSSIONS

This subsection provides additional analyses of our proposed FedTRL, including its generalization to unseen domains, its representation- and gradient-level behavior in handling heterogeneity, and its robustness under different simulated heterogeneity settings.

**Generalization to Unseen Domain.** To further test generalization, we conducted a leave-one-domain-out study (Tables 18–19 below), where an entire domain was removed during pretraining.

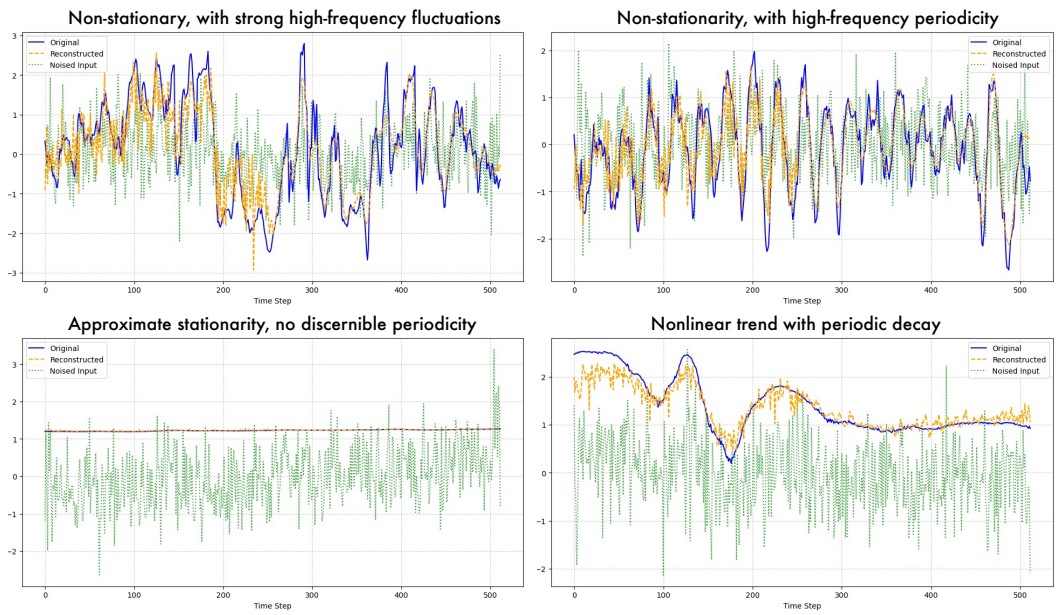

Figure 8: Showcase of diffusion-based reconstruction during pretraining.

Even under this stricter setting, FedTRL consistently outperforms centralized mixed-domain pre-training and other FL baselines on the unseen domain. Performance naturally decreases when excluding related domains (an expected phenomenon for all foundation models) but FedTRL remains the most robust under such conditions.

**Representation- and Gradient-Level Analysis.** Table 20 shows that centralized mixed-domain pretraining yields the smallest inter-domain centroid distance and the largest within-domain variance, indicating collapsed and weakly structured representations. In contrast, FedTRL maintains clear domain separation while producing the most compact within-domain clusters. Table 21 further shows that All-Mixed exhibits the strongest gradient conflict (lowest cosine similarity & highest gradient-norm variance), whereas FedTRL substantially alleviates these conflicts.

**Robustness Under Increasing Heterogeneity.** By gradually increasing domain imbalance (from Balanced to Mildly Imbalanced to Real Skew) through controlling the data scale of each domain, we observe consistent performance degradation for All-Mixed, FedAvg, and FFTS across in-domain, full-shot, zero-shot, and probabilistic forecasting (as shown in Table 22). In contrast, FedTRL remains markedly more stable in all settings, indicating that severe heterogeneity indeed harms federated training and that FedTRL effectively mitigates this effect.

### C.8 Showcases

**Reconstruction from Noise in Pretraining.** To illustrate the effectiveness of our diffusion-based reconstruction strategy, we showcase examples of noisy inputs, reconstructions, and original sequences. As shown in Fig. 8, the model successfully restores a wide range of temporal dynamics, including non-stationary series with high-frequency fluctuations, periodic components, approximate stationarity, and nonlinear trends. These cases highlight the ability of FedTRL to preserve fine-grained temporal structures and denoise corrupted inputs, even under complex and unstable conditions, thereby ensuring robust representation learning during pretraining.

**Forecasting.** Fig. 9- 10 present zero-shot forecasting showcases on all the datasets from our RW-Bench (point forecasting) and FEV-leaderboard (Ansari et al., 2024) (probabilistic forecasting).

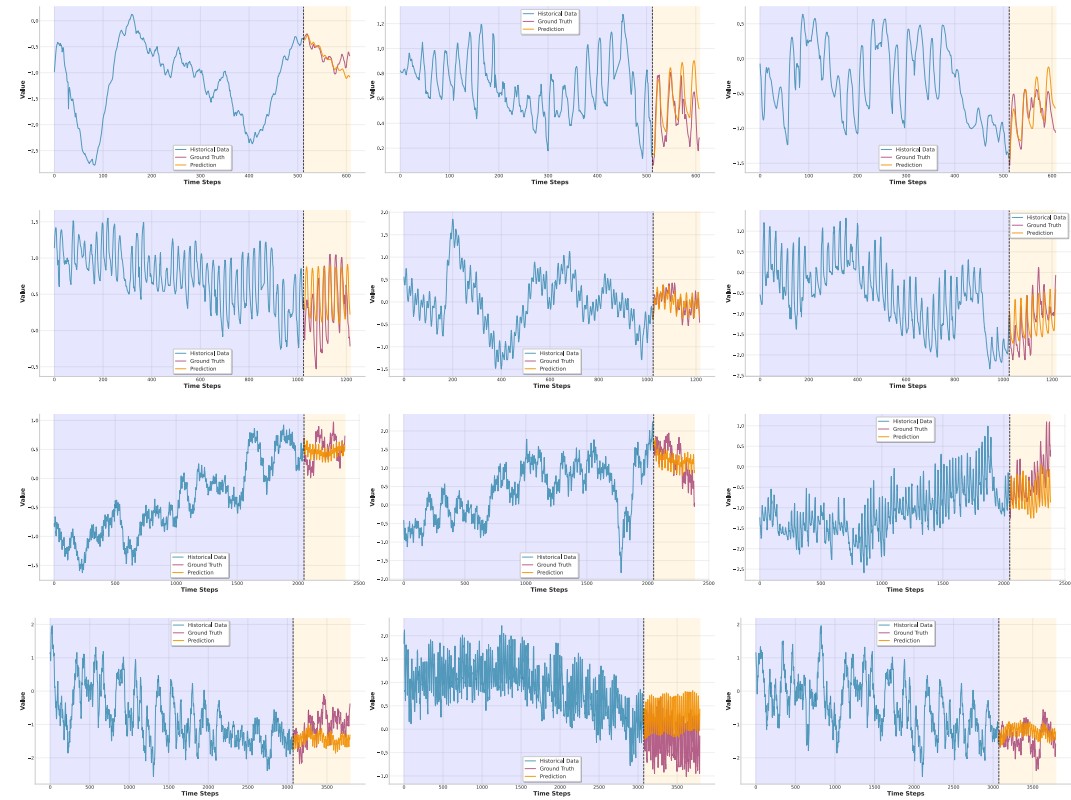

Figure 9: Zero-shot point forecasting results on our RW-Bench. From top to bottom, the prediction horizons are 96, 192, 336, and 720, corresponding to lookback window lengths of 512, 1024, 2048, and 3072, respectively. The visual samples are randomly selected from RW-Bench.

## D  BROADER IMPACTS

FedTRL introduces a federated strategy for training time series foundation models without centralizing raw data. This design can benefit real-world domains such as climate, energy, healthcare, finance, and urban management by providing more reliable and generalizable forecasting under heterogeneous conditions. By supporting decentralized pretraining, FedTRL reduces risks associated with data silos and privacy concerns, while promoting collaboration across organizations that cannot openly share sensitive data. Beyond technical contributions, FedTRL represents a potential paradigm shift: moving from centralized collection to decentralized collaboration for building large-scale models. This approach may broaden access to foundation-level forecasting tools and foster more trustworthy decision-making in dynamic, high-stakes environments.

## THE USE OF LARGE LANGUAGE MODELS

We declare that in the preparation of this paper, we utilized a Large Language Model (ChatGPT 5) as a general-purpose assist tool. Its primary use was for language polishing and grammar correction, including improving sentence flow, enhancing vocabulary selection, and ensuring the text adheres to academic writing standards. All core research work, including ideation, experimental design, data analysis, and conclusion derivation, was performed independently by the authors. The LLM was not used for any substantive content generation. We take full responsibility for all content within this paper and guarantee its authenticity and originality.

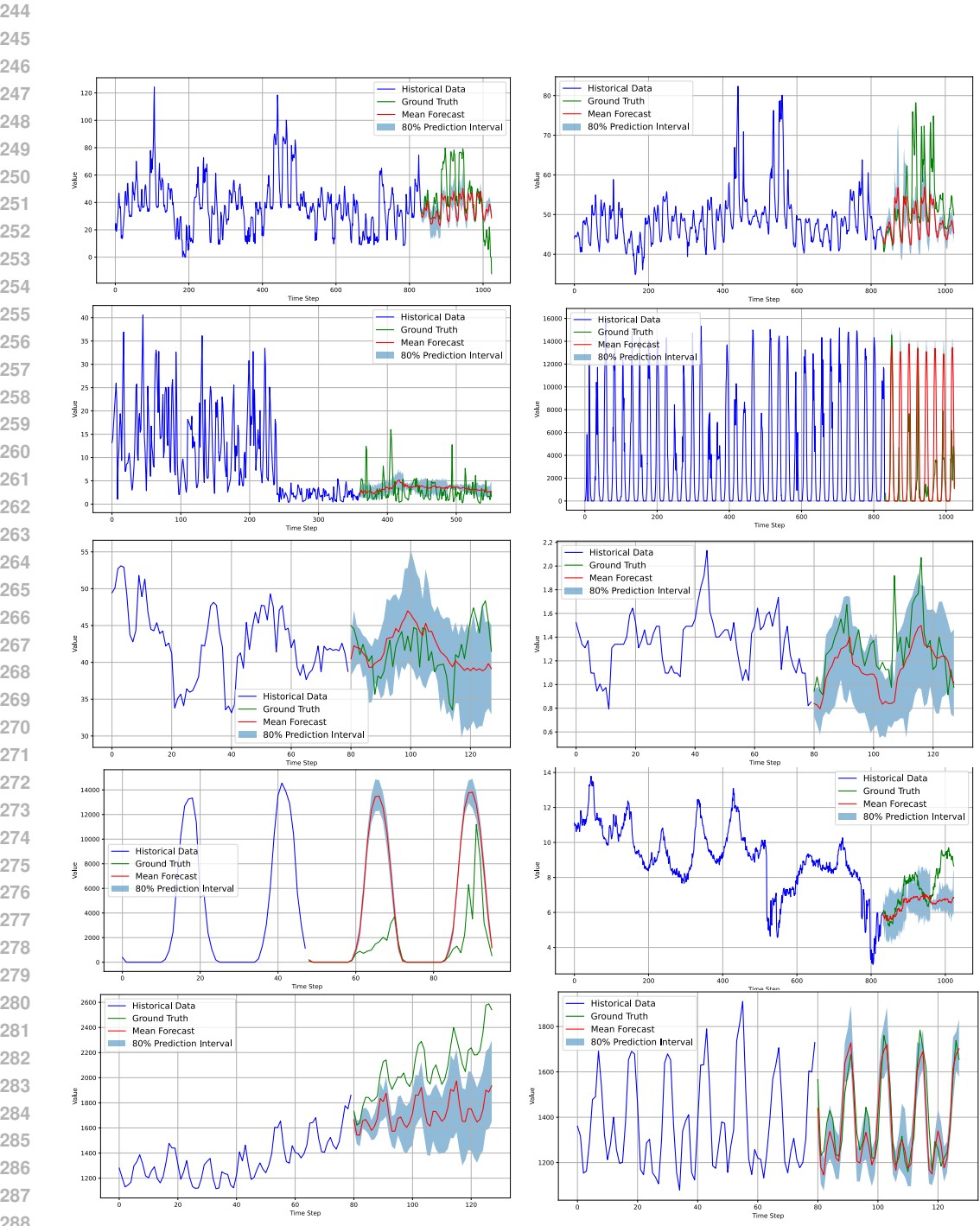

Figure 10: Zero-shot probabilistic forecasting results on FEV-leaderboard datasets, with visual samples randomly drawn from the benchmark.

Table 14: Full results of real-world weather benchmark (RW-bench).

| Models | Metrics | Ours FedTRL | | Time-MoE$_{base}$ | | Time-MoE$_{large}$ | | Moirai$_{small}$ | | Moirai$_{base}$ | | Moirai$_{large}$ | | TimesFM | | Moment | | Chronos$_{small}$ | | Chronos$_{base}$ | | Chronos$_{large}$ | |
|---|---|---|---|---|---|---|---|---|---|---|---|---|---|---|---|---|---|---|---|---|---|---|---|
| | | MSE | MAE | MSE | MAE | MSE | MAE | MSE | MAE | MSE | MAE | MSE | MAE | MSE | MAE | MSE | MAE | MSE | MAE | MSE | MAE | MSE | MAE |
| RW1 | 96 | 0.508 | 0.453 | 0.510 | 0.460 | 0.505 | 0.455 | 0.885 | 0.547 | 0.879 | 0.529 | 0.986 | 0.530 | 0.883 | 0.703 | 0.674 | 0.594 | 0.850 | 0.581 | 0.879 | 0.589 | 0.873 | 0.583 |
| | 192 | 0.528 | 0.473 | 0.540 | 0.492 | 0.532 | 0.485 | 0.950 | 0.569 | 0.896 | 0.546 | 1.011 | 0.548 | 0.903 | 0.711 | 0.717 | 0.600 | 0.917 | 0.620 | 0.979 | 0.634 | 0.950 | 0.623 |
| | 336 | 0.546 | 0.488 | 0.566 | 0.519 | 0.556 | 0.505 | 1.148 | 0.588 | 0.813 | 0.552 | 0.868 | 0.560 | 0.889 | 0.708 | 0.913 | 0.700 | 0.957 | 0.649 | 1.047 | 0.667 | 0.973 | 0.646 |
| | 720 | 0.552 | 0.497 | 0.643 | 0.580 | 0.568 | 0.519 | 1.869 | 0.667 | 0.958 | 0.602 | 1.259 | 0.625 | 0.877 | 0.705 | 1.740 | 1.019 | 1.089 | 0.708 | 1.142 | 0.715 | 1.012 | 0.687 |
| | Avg | 0.534 | 0.477 | 0.565 | 0.513 | 0.540 | 0.491 | 1.213 | 0.593 | 0.887 | 0.557 | 1.031 | 0.566 | 0.888 | 0.707 | 1.011 | 0.728 | 0.953 | 0.639 | 1.012 | 0.651 | 0.952 | 0.635 |
| RW2 | 96 | 0.433 | 0.420 | 0.431 | 0.426 | 0.436 | 0.424 | 1.541 | 0.526 | 1.314 | 0.483 | 1.877 | 0.491 | 0.846 | 0.702 | 0.677 | 0.567 | 0.778 | 0.543 | 0.779 | 0.546 | 0.811 | 0.553 |
| | 192 | 0.460 | 0.445 | 0.460 | 0.455 | 0.470 | 0.456 | 1.407 | 0.560 | 1.021 | 0.504 | 2.416 | 0.529 | 0.841 | 0.700 | 0.788 | 0.623 | 0.855 | 0.587 | 0.874 | 0.595 | 0.895 | 0.598 |
| | 336 | 0.481 | 0.459 | 0.497 | 0.487 | 0.497 | 0.477 | 1.666 | 0.596 | 1.168 | 0.529 | 1.910 | 0.553 | 0.842 | 0.700 | 0.893 | 0.625 | 0.942 | 0.629 | 0.960 | 0.630 | | |
| | 720 | 0.505 | 0.481 | 0.575 | 0.545 | 0.539 | 0.505 | 2.236 | 0.660 | 2.600 | 0.603 | 4.302 | 0.637 | 0.847 | 0.702 | 1.933 | 1.044 | 1.146 | 0.691 | 1.054 | 0.681 | 1.049 | 0.677 |
| | Avg | 0.470 | 0.451 | 0.491 | 0.479 | 0.485 | 0.465 | 1.713 | 0.585 | 1.526 | 0.530 | 2.626 | 0.552 | 0.844 | 0.701 | 1.070 | 0.721 | 0.931 | 0.612 | 0.912 | 0.613 | 0.929 | 0.614 |
| RW3 | 96 | 0.580 | 0.485 | 0.566 | 0.474 | 0.579 | 0.485 | 1.461 | 0.616 | 1.254 | 0.576 | 1.511 | 0.561 | 0.952 | 0.728 | 0.726 | 0.598 | 0.977 | 0.614 | 0.980 | 0.625 | 0.943 | 0.600 |
| | 192 | 0.594 | 0.501 | 0.594 | 0.498 | 0.597 | 0.508 | 1.224 | 0.610 | 0.991 | 0.577 | 1.450 | 0.571 | 0.963 | 0.730 | 0.808 | 0.629 | 1.032 | 0.649 | 1.058 | 0.668 | 1.029 | 0.640 |
| | 336 | 0.605 | 0.511 | 0.620 | 0.522 | 0.608 | 0.519 | 1.578 | 0.629 | 0.900 | 0.583 | 1.053 | 0.587 | 0.973 | 0.735 | 0.908 | 0.674 | 1.084 | 0.675 | 1.121 | 0.696 | 1.054 | 0.658 |
| | 720 | 0.617 | 0.529 | 0.681 | 0.576 | 0.631 | 0.544 | 2.740 | 0.709 | 1.233 | 0.650 | 1.755 | 0.653 | 0.953 | 0.732 | 2.197 | 1.137 | 1.181 | 0.725 | 1.218 | 0.741 | 1.165 | 0.729 |
| | Avg | 0.599 | 0.506 | 0.615 | 0.517 | 0.604 | 0.514 | 1.751 | 0.641 | 1.094 | 0.597 | 1.442 | 0.593 | 0.960 | 0.731 | 1.159 | 0.760 | 1.068 | 0.666 | 1.094 | 0.682 | 1.048 | 0.657 |
| RW4 | 96 | 0.501 | 0.484 | 0.512 | 0.486 | 0.502 | 0.483 | 1.634 | 0.603 | 1.313 | 0.562 | 1.808 | 0.571 | 0.895 | 0.728 | 0.707 | 0.588 | 0.891 | 0.629 | 0.870 | 0.628 | 0.889 | 0.630 |
| | 192 | 0.517 | 0.501 | 0.535 | 0.510 | 0.522 | 0.505 | 1.299 | 0.606 | 1.104 | 0.571 | 1.651 | 0.592 | 0.886 | 0.726 | 0.776 | 0.619 | 0.951 | 0.663 | 0.959 | 0.671 | 0.961 | 0.669 |
| | 336 | 0.534 | 0.512 | 0.557 | 0.534 | 0.544 | 0.522 | 1.457 | 0.629 | 0.931 | 0.575 | 1.174 | 0.610 | 0.893 | 0.727 | 0.862 | 0.667 | 1.004 | 0.690 | 1.025 | 0.701 | 1.013 | 0.695 |
| | 720 | 0.558 | 0.530 | 0.616 | 0.582 | 0.582 | 0.547 | 2.364 | 0.703 | 1.576 | 0.644 | 2.106 | 0.683 | 0.888 | 0.726 | 2.205 | 1.126 | 1.169 | 0.747 | 1.170 | 0.757 | 1.089 | 0.740 |
| | Avg | 0.527 | 0.507 | 0.555 | 0.528 | 0.538 | 0.514 | 1.688 | 0.635 | 1.231 | 0.588 | 1.685 | 0.614 | 0.890 | 0.727 | 1.138 | 0.750 | 1.004 | 0.682 | 1.006 | 0.689 | 0.988 | 0.684 |
| RW5 | 96 | 0.489 | 0.469 | 0.491 | 0.470 | 0.498 | 0.468 | 1.002 | 0.571 | 0.911 | 0.541 | 0.941 | 0.533 | 0.900 | 0.732 | 0.707 | 0.600 | 0.897 | 0.622 | 0.911 | 0.626 | 0.924 | 0.627 |
| | 192 | 0.509 | 0.490 | 0.513 | 0.494 | 0.529 | 0.491 | 1.049 | 0.588 | 0.815 | 0.547 | 0.912 | 0.551 | 0.912 | 0.735 | 0.759 | 0.617 | 0.973 | 0.661 | 0.998 | 0.672 | 0.991 | 0.668 |
| | 336 | 0.517 | 0.499 | 0.538 | 0.516 | 0.529 | 0.507 | 1.503 | 0.614 | 0.805 | 0.555 | 1.040 | 0.577 | 0.905 | 0.732 | 1.013 | 0.727 | 1.039 | 0.695 | 1.071 | 0.706 | 1.053 | 0.696 |
| | 720 | 0.526 | 0.510 | 0.613 | 0.570 | 0.541 | 0.524 | 2.697 | 0.687 | 1.226 | 0.616 | 1.825 | 0.653 | 0.910 | 0.735 | 1.741 | 1.008 | 1.155 | 0.748 | 1.199 | 0.755 | 1.166 | 0.738 |
| | Avg | 0.510 | 0.492 | 0.539 | 0.512 | 0.520 | 0.497 | 1.563 | 0.615 | 0.939 | 0.565 | 1.167 | 0.578 | 0.907 | 0.733 | 1.055 | 0.738 | 1.016 | 0.682 | 1.045 | 0.690 | 1.033 | 0.683 |
| RW6 | 96 | 0.786 | 0.499 | 0.796 | 0.516 | 0.785 | 0.499 | 1.442 | 0.596 | 1.169 | 0.566 | 1.555 | 0.566 | 1.290 | 0.778 | 0.692 | 0.597 | 1.365 | 0.656 | 1.388 | 0.660 | 1.349 | 0.660 |
| | 192 | 0.808 | 0.517 | 0.808 | 0.531 | 0.809 | 0.521 | 1.228 | 0.597 | 1.021 | 0.565 | 1.706 | 0.586 | 1.299 | 0.779 | 0.754 | 0.627 | 1.462 | 0.692 | 1.533 | 0.704 | 1.463 | 0.699 |
| | 336 | 0.821 | 0.525 | 0.835 | 0.549 | 0.828 | 0.537 | 1.733 | 0.623 | 1.126 | 0.585 | 1.456 | 0.610 | 1.301 | 0.780 | 0.860 | 0.667 | 1.524 | 0.720 | 1.630 | 0.737 | 1.552 | 0.727 |
| | 720 | 0.831 | 0.539 | 0.897 | 0.599 | 0.837 | 0.552 | 2.410 | 0.694 | 1.877 | 0.657 | 2.881 | 0.690 | 1.290 | 0.780 | 1.703 | 0.982 | 1.627 | 0.772 | 1.752 | 0.786 | 1.670 | 0.766 |
| | Avg | 0.812 | 0.520 | 0.834 | 0.549 | 0.815 | 0.527 | 1.703 | 0.628 | 1.298 | 0.592 | 1.900 | 0.613 | 1.295 | 0.779 | 1.002 | 0.718 | 1.495 | 0.710 | 1.576 | 0.722 | 1.508 | 0.713 |
| RW7 | 96 | 0.479 | 0.478 | 0.492 | 0.481 | 0.493 | 0.479 | 0.703 | 0.550 | 0.695 | 0.541 | 0.660 | 0.531 | 0.887 | 0.726 | 0.728 | 0.615 | 0.879 | 0.625 | 0.897 | 0.626 | 0.888 | 0.626 |
| | 192 | 0.503 | 0.504 | 0.514 | 0.503 | 0.514 | 0.504 | 0.712 | 0.561 | 0.686 | 0.549 | 0.655 | 0.553 | 0.891 | 0.727 | 0.817 | 0.634 | 0.939 | 0.662 | 0.981 | 0.670 | 0.953 | 0.665 |
| | 336 | 0.518 | 0.520 | 0.536 | 0.525 | 0.527 | 0.522 | 0.791 | 0.582 | 0.696 | 0.567 | 0.661 | 0.566 | 0.885 | 0.725 | 0.899 | 0.696 | 0.981 | 0.687 | 1.028 | 0.697 | 1.001 | 0.691 |
| | 720 | 0.526 | 0.532 | 0.582 | 0.564 | 0.537 | 0.533 | 1.462 | 0.643 | 0.733 | 0.610 | 0.868 | 0.615 | 0.880 | 0.724 | 1.961 | 1.071 | 1.081 | 0.731 | 1.094 | 0.735 | 1.021 | 0.722 |
| | Avg | 0.506 | 0.508 | 0.531 | 0.518 | 0.518 | 0.509 | 0.917 | 0.584 | 0.703 | 0.567 | 0.711 | 0.566 | 0.886 | 0.725 | 1.101 | 0.754 | 0.970 | 0.676 | 1.000 | 0.682 | 0.966 | 0.676 |
| RW8 | 96 | 0.520 | 0.468 | 0.525 | 0.471 | 0.527 | 0.466 | 0.853 | 0.552 | 0.828 | 0.525 | 0.881 | 0.531 | 1.035 | 0.748 | 0.786 | 0.643 | 0.858 | 0.598 | 0.929 | 0.622 | 0.910 | 0.615 |
| | 192 | 0.539 | 0.492 | 0.538 | 0.489 | 0.542 | 0.489 | 0.769 | 0.563 | 0.773 | 0.539 | 0.736 | 0.546 | 1.019 | 0.746 | 0.920 | 0.701 | 0.925 | 0.638 | 1.058 | 0.679 | 1.012 | 0.665 |
| | 336 | 0.545 | 0.503 | 0.557 | 0.509 | 0.573 | 0.502 | 0.800 | 0.581 | 0.805 | 0.557 | 0.718 | 0.559 | 1.018 | 0.748 | 1.026 | 0.752 | 1.026 | 0.668 | 1.170 | 0.723 | 1.119 | 0.702 |
| | 720 | 0.560 | 0.516 | 0.621 | 0.556 | 0.573 | 0.527 | 1.054 | 0.631 | 0.818 | 0.589 | 0.937 | 0.614 | 1.018 | 0.748 | 2.017 | 1.111 | 1.147 | 0.718 | 1.356 | 0.789 | 1.298 | 0.777 |
| | Avg | 0.541 | 0.495 | 0.560 | 0.506 | 0.547 | 0.496 | 0.869 | 0.582 | 0.806 | 0.553 | 0.818 | 0.563 | 1.022 | 0.748 | 1.187 | 0.802 | 0.979 | 0.655 | 1.128 | 0.703 | 1.085 | 0.690 |
| RW9 | 96 | 0.576 | 0.479 | 0.576 | 0.476 | 0.585 | 0.482 | 1.143 | 0.578 | 1.005 | 0.557 | 1.304 | 0.559 | 1.012 | 0.720 | 0.791 | 0.645 | 0.939 | 0.605 | 1.062 | 0.636 | 1.041 | 0.630 |
| | 192 | 0.622 | 0.513 | 0.613 | 0.508 | 0.629 | 0.517 | 1.124 | 0.607 | 0.996 | 0.582 | 1.066 | 0.586 | 1.025 | 0.725 | 0.913 | 0.685 | 1.039 | 0.654 | 1.228 | 0.703 | 1.178 | 0.695 |
| | 336 | 0.644 | 0.528 | 0.648 | 0.532 | 0.648 | 0.529 | 1.121 | 0.623 | 0.966 | 0.596 | 1.061 | 0.599 | 1.036 | 0.728 | 1.091 | 0.791 | 1.106 | 0.686 | 1.357 | 0.748 | 1.321 | 0.738 |
| | 720 | 0.658 | 0.544 | 0.707 | 0.573 | 0.669 | 0.552 | 1.313 | 0.661 | 1.029 | 0.634 | 1.557 | 0.704 | 1.037 | 0.731 | 2.049 | 1.133 | 1.221 | 0.734 | 1.518 | 0.802 | 1.500 | 0.778 |
| | Avg | 0.625 | 0.516 | 0.636 | 0.522 | 0.633 | 0.520 | 1.175 | 0.617 | 0.999 | 0.592 | 1.247 | 0.617 | 1.028 | 0.726 | 1.211 | 0.813 | 1.076 | 0.670 | 1.291 | 0.722 | 1.260 | 0.710 |
| RW10 | 96 | 0.566 | 0.505 | 0.585 | 0.518 | 0.579 | 0.505 | 0.923 | 0.607 | 0.926 | 0.591 | 0.836 | 0.578 | 0.989 | 0.768 | 0.752 | 0.626 | 1.014 | 0.657 | 1.073 | 0.661 | 1.016 | 0.651 |
| | 192 | 0.599 | 0.537 | 0.612 | 0.546 | 0.610 | 0.537 | 0.935 | 0.629 | 0.949 | 0.617 | 0.874 | 0.615 | 0.986 | 0.768 | 0.777 | 0.638 | 1.087 | 0.703 | 1.144 | 0.708 | 1.098 | 0.698 |
| | 336 | 0.616 | 0.553 | 0.626 | 0.566 | 0.621 | 0.554 | 0.907 | 0.637 | 0.972 | 0.633 | 0.864 | 0.634 | 0.976 | 0.768 | 0.852 | 0.678 | 1.117 | 0.728 | 1.173 | 0.730 | 1.120 | 0.720 |
| | 720 | 0.618 | 0.564 | 0.655 | 0.600 | 0.629 | 0.564 | 1.119 | 0.682 | 1.063 | 0.680 | 1.070 | 0.690 | 0.968 | 0.767 | 1.872 | 1.069 | 1.206 | 0.774 | 1.260 | 0.774 | 1.186 | 0.755 |
| | Avg | 0.600 | 0.540 | 0.620 | 0.558 | 0.609 | 0.540 | 0.971 | 0.639 | 0.977 | 0.630 | 0.911 | 0.629 | 0.980 | 0.768 | 1.063 | 0.753 | 1.106 | 0.716 | 1.162 | 0.718 | 1.105 | 0.706 |
| RW11 | 96 | 0.529 | 0.487 | 0.556 | 0.498 | 0.546 | 0.495 | 0.975 | 0.601 | 0.929 | 0.593 | 0.967 | 0.583 | 0.901 | 0.740 | 0.707 | 0.623 | 0.906 | 0.601 | 0.956 | 0.608 | 0.906 | 0.596 |
| | 192 | 0.586 | 0.539 | 0.588 | 0.535 | 0.596 | 0.542 | 1.062 | 0.642 | 0.999 | 0.630 | 0.936 | 0.619 | 0.898 | 0.738 | 0.726 | 0.625 | 0.999 | 0.656 | 1.108 | 0.670 | 1.014 | 0.651 |
| | 336 | 0.628 | 0.567 | 0.614 | 0.559 | 0.639 | 0.571 | 0.916 | 0.635 | 0.900 | 0.643 | 0.846 | 0.635 | 0.906 | 0.740 | 0.985 | 0.765 | 1.089 | 0.690 | 1.146 | 0.698 | 1.070 | 0.679 |
| | 720 | 0.655 | 0.586 | 0.673 | 0.600 | 0.655 | 0.585 | 0.931 | 0.669 | 0.915 | 0.675 | 0.920 | 0.675 | 0.908 | 0.739 | 1.685 | 1.023 | 1.264 | 0.738 | 1.267 | 0.746 | 1.328 | 0.712 |
| | Avg | 0.599 | 0.545 | 0.608 | 0.548 | 0.609 | 0.548 | 0.971 | 0.635 | 0.936 | 0.635 | 0.917 | 0.628 | 0.903 | 0.739 | 1.026 | 0.759 | 1.065 | 0.671 | 1.119 | 0.681 | 1.080 | 0.659 |
| RW12 | 96 | 0.708 | 0.554 | 0.751 | 0.569 | 0.726 | 0.556 | 1.124 | 0.661 | 1.168 | 0.657 | 1.134 | 0.641 | 1.140 | 0.798 | 0.782 | 0.654 | 1.196 | 0.693 | 1.227 | 0.684 | 1.191 | 0.682 |
| | 192 | 0.711 | 0.605 | 0.791 | 0.605 | 0.781 | 0.606 | 1.285 | 0.707 | 1.206 | 0.690 | 1.183 | 0.688 | 1.149 | 0.802 | 0.836 | 0.650 | 1.331 | 0.754 | 1.415 | 0.755 | 1.342 | 0.746 |
| | 336 | 0.811 | 0.629 | 0.821 | 0.631 | 0.827 | 0.637 | 1.544 | 0.720 | 1.252 | 0.720 | 1.236 | 0.722 | 1.146 | 0.800 | 1.011 | 0.755 | 1.423 | 0.792 | 1.502 | 0.793 | 1.454 | 0.783 |
| | 720 | 0.838 | 0.650 | 0.880 | 0.668 | 0.858 | 0.657 | 2.417 | 0.780 | 1.342 | 0.768 | 1.529 | 0.730 | 1.158 | 0.804 | 1.910 | 1.092 | 1.589 | 0.842 | 1.662 | 0.846 | 1.572 | 0.827 |
| | Avg | 0.782 | 0.610 | 0.811 | 0.618 | 0.798 | 0.614 | 1.592 | 0.717 | 1.242 | 0.709 | 1.270 | 0.710 | 1.148 | 0.801 | 1.135 | 0.802 | 1.385 | 0.770 | 1.451 | 0.770 | 1.390 | 0.759 |
| RW13 | 96 | 0.584 | 0.493 | 0.601 | 0.509 | 0.597 | 0.499 | 0.969 | 0.593 | 0.832 | 0.571 | 0.845 | 0.570 | 1.047 | 0.773 | 0.736 | 0.623 | 1.057 | 0.657 | 1.081 | 0.650 | 1.075 | 0.650 |
| | 192 | 0.619 | 0.522 | 0.643 | 0.541 | 0.622 | 0.523 | 1.055 | 0.616 | 0.908 | 0.601 | 0.944 | 0.609 | 1.045 | 0.770 | 0.781 | 0.636 | 1.149 | 0.706 | 1.202 | 0.701 | 1.195 | 0.693 |
| | 336 | 0.644 | 0.539 | 0.668 | 0.564 | 0.648 | 0.542 | 0.986 | 0.632 | 0.901 | 0.617 | 0.914 | 0.627 | 1.048 | 0.771 | 0.901 | 0.688 | 1.232 | 0.740 | 1.248 | 0.732 | 1.195 | 0.723 |
| | 720 | 0.657 | 0.556 | 0.725 | 0.613 | 0.660 | 0.556 | 1.290 | 0.689 | 0.921 | 0.654 | 1.021 | 0.666 | 1.035 | 0.771 | 1.925 | 1.065 | 1.329 | 0.793 | 1.299 | 0.776 | 1.243 | 0.755 |
| | Avg | 0.626 | 0.528 | 0.659 | 0.557 | 0.632 | 0.530 | 1.075 | 0.633 | 0.891 | 0.611 | 0.931 | 0.618 | 1.044 | 0.771 | 1.086 | 0.753 | 1.192 | 0.724 | 1.208 | 0.715 | 1.166 | 0.705 |
| RW14 | 96 | 0.648 | 0.520 | 0.666 | 0.532 | 0.653 | 0.525 | 1.144 | 0.620 | 1.044 | 0.602 | 1.111 | 0.597 | 1.107 | 0.782 | 0.743 | 0.630 | 1.118 | 0.667 | 1.166 | 0.674 | 1.129 | 0.669 |
| | 192 | 0.682 | 0.550 | 0.696 | 0.560 | 0.688 | 0.557 | 1.211 | 0.646 | 0.989 | 0.615 | 1.026 | 0.623 | 1.098 | 0.777 | 0.819 | 0.649 | 1.221 | 0.713 | 1.281 | 0.727 | 1.256 | 0.721 |
| | 336 | 0.714 | 0.573 | 0.722 | 0.581 | 0.720 | 0.580 | 1.358 | 0.674 | 1.029 | 0.637 | 1.080 | 0.648 | 1.109 | 0.780 | 0.883 | 0.688 | 1.277 | 0.742 | 1.401 | 0.766 | 1.314 | 0.752 |
| | 720 | 0.724 | 0.587 | 0.789 | 0.628 | 0.734 | 0.592 | 1.793 | 0.732 | 1.096 | 0.689 | 1.219 | 0.697 | 1.099 | 0.778 | 1.891 | 1.067 | 1.359 | 0.786 | 1.459 | 0.805 | 1.389 | 0.783 |
| | Avg | 0.692 | 0.557 | 0.718 | 0.575 | 0.699 | 0.563 | 1.376 | 0.668 | 1.039 | 0.636 | 1.109 | 0.641 | 1.103 | 0.780 | 1.084 | 0.759 | 1.244 | 0.727 | 1.327 | 0.743 | 1.272 | 0.731 |
| RW15 | 96 | 0.508 | 0.484 | 0.505 | 0.483 | 0.512 | 0.488 | 0.891 | 0.565 | 0.797 | 0.559 | 0.765 | 0.540 | 0.972 | 0.760 | 0.740 | 0.623 | 0.843 | 0.587 | 0.861 | 0.600 | 0.836 | 0.588 |
| | 192 | 0.559 | 0.524 | 0.547 | 0.519 | 0.559 | 0.529 | 1.051 | 0.613 | 0.837 | 0.599 | 0.837 | 0.584 | 0.962 | 0.758 | 0.751 | 0.632 | 0.932 | 0.635 | 0.980 | 0.659 | 0.933 | 0.639 |
| | 336 | 0.583 | 0.543 | 0.595 | 0.555 | 0.594 | 0.555 | 1.329 | 0.638 | 0.852 | 0.615 | 0.886 | 0.616 | 0.973 | 0.762 | 0.852 | 0.689 | 0.999 | 0.664 | 1.045 | 0.693 | 0.998 | 0.670 |
| | 720 | 0.616 | 0.568 | 0.672 | 0.608 | 0.641 | 0.585 | 2.119 | 0.704 | 0.967 | 0.663 | 1.145 | 0.682 | 0.976 | 0.762 | 1.787 | 1.044 | 1.120 | 0.715 | 1.169 | 0.745 | 1.101 | 0.740 |
| | Avg | 0.566 | 0.530 | 0.580 | 0.541 | 0.576 | 0.539 | 1.348 | 0.630 | 0.863 | 0.609 | 0.888 | 0.606 | 0.971 | 0.760 | 1.033 | 0.747 | 0.974 | 0.650 | 1.014 | 0.675 | 0.967 | 0.659 |
| 1$^{st}$ Count | | 124 | | 21 | | 12 | | 0 | | 0 | | 0 | | 0 | | 2 | | 0 | | 0 | | 0 | |

Table 15: Additional zero-shot forecasting results averaged over four horizons (96, 192, 336, 720).

| Dataset/Model | FedTRL (Ours) | VisionTS | Sundial (94B) | Sundial (230B) | Sundial (1032B) |
|---|---|---|---|---|---|
| ETTh1 | **0.399** | 0.412 | 0.402 | 0.403 | 0.411 |
| ETTh2 | **0.350** | 0.365 | 0.377 | 0.364 | 0.333 |
| ETTm1 | **0.349** | 0.391 | 0.367 | 0.352 | 0.336 |
| ETTm2 | **0.282** | 0.296 | 0.280 | 0.273 | 0.258 |
| Weather | **0.238** | 0.303 | 0.254 | 0.252 | 0.234 |
| RW-Bench | **0.599** | 0.727 | 0.618 | 0.617 | 0.602 |
| $1^{st}$ Count | **6** | 0 | 0 | 0 | 0 |

Table 16: Model configurations of scalability exploration, where gray shading is the default setting.

| Patch Size $(P)$ | Context Length $(T)$ | Prediction Length $(F)$ | Layers $(L)$ | Dimension $(D, D_{\text{ff}})$ | Heads $H$ | Total Parameters #Count |
|---|---|---|---|---|---|---|
| 16 | 3072 | $\{16, 720\}$ | 12 | $(512, 2048)$ | 8 | 38M |
| 16 | 3072 | $\{16, 720\}$ | 16 | $(768, 3072)$ | 12 | 114M |
| 16 | 3072 | $\{16, 720\}$ | 24 | $(1024, 4096)$ | 16 | 302M |

Table 17: Full/Zero-shot point forecasting (MSE report) and zero-shot probabilistic forecasting (Avg. MASE report) results across different pretraining datasets scales.

| Task / Report | Datasets / Metric | Pretraining Datasets Scale (Billion) | | | | | |
|---|---|---|---|---|---|---|---|
| | | 90B | 120B | 180B | 300B (Original) | 450B | 540B |
| **Full-shot Forecasting** | | | | | | | |
| MSE Report | ETTh1 | 0.401 | 0.387 | 0.372 | 0.371 | 0.363 | **0.363** |
| | ETTm1 | 0.343 | 0.336 | 0.328 | 0.316 | 0.312 | **0.310** |
| | Weather | 0.232 | 0.231 | 0.227 | 0.214 | 0.204 | **0.202** |
| **Zero-shot Point Forecasting** | | | | | | | |
| MSE Report | ETTh1 | 0.428 | 0.432 | 0.411 | 0.399 | 0.400 | **0.393** |
| | ETTm1 | 0.402 | 0.371 | 0.376 | 0.350 | 0.344 | **0.341** |
| | Weather | 0.275 | 0.273 | 0.269 | 0.238 | 0.214 | **0.208** |
| | RW-Bench | 0.689 | 0.647 | 0.641 | 0.599 | 0.581 | **0.575** |
| **Zero-shot Probabilistic** | | | | | | | |
| Ave. MASE Report | GIFT-eval | 0.770 | 0.744 | 0.721 | 0.675 | 0.677 | **0.669** |
| | FEV Leaderboard | 0.945 | 0.873 | 0.847 | 0.836 | 0.833 | **0.829** |

Table 18: In-domain forecasting results (MSE averaged across $\{96, 192, 336, 720\}$). Corresponding domain dataset has been excluded from the pretraining dataset (energy, traffic, respectively).

| Method | ETTh1 | ETTm1 | Traffic | Weather |
|---|---|---|---|---|
| FedAvg | 0.511 | 0.433 | 0.488 | 0.295 |
| FFTS | 0.492 | 0.421 | 0.479 | 0.290 |
| Centralized All Mixed | 0.498 | 0.427 | 0.483 | 0.290 |
| FedTRL (Original) | 0.448 | 0.375 | 0.426 | 0.241 |
| **FedTRL (Ours)** | **0.472** | **0.411** | **0.477** | **0.282** |

Table 19: Zero-shot forecasting results (MSE averaged across $\{96, 192, 336, 720\}$). Corresponding domain dataset has been excluded from the pretraining dataset (energy, traffic, respectively).

| Method | ETTh1 | ETTm1 | Weather |
|---|---|---|---|
| FedAvg | 0.438 | 0.384 | 0.292 |
| FFTS | 0.427 | 0.380 | 0.282 |
| Centralized All Mixed | 0.431 | 0.382 | 0.288 |
| FedTRL (Original) | 0.399 | 0.349 | 0.238 |
| **FedTRL** | 0.418 ($\downarrow 4.76\%$) | 0.366 ($\downarrow 4.87\%$) | 0.269 ($\downarrow 13.03\%$) |

Table 20: Representation-level Domain Analysis. We report the average inter-domain centroid distance (higher is better separation) and average within-domain variance (lower is more compact).

| Method | Inter-domain Centroid Distance ($\uparrow$) | Within-domain Variance ($\downarrow$) |
|---|---|---|
| All Mixed (Centralized) | 1.08 | 0.91 |
| FedAvg | 1.06 | 0.85 |
| FFTS | 1.34 | 0.69 |
| **FedTRL (Ours)** | **1.27** | **0.53** |

Table 21: Gradient Conflict Analysis Across Domains. We report the average pairwise cosine similarity between domain-wise gradients (higher means less conflict) and the variance of gradient norms across domains (lower means more balanced contributions).

| Method | Avg. Gradient Cosine Similarity ($\uparrow$) | Gradient Norm Variance ($\downarrow$) |
|---|---|---|
| All Mixed (Centralized) | 0.23 | 0.021 |
| FedAvg | 0.33 | 0.015 |
| FFTS | 0.37 | 0.013 |
| **FedTRL (Ours)** | **0.43** | **0.010** |

Table 22: Forecasting results under different inter-domain imbalance levels.

| Task Type / Dataset (Metric) | Method | Inter-domain Imbalance Level | | |
|---|---|---|---|---|
| | | Balanced (Simulated) | Mildly Imbalanced (Simulated) | Real Skew (Original Pretraining) |
| **In-domain Forecasting** (MSE on Weather) | All Mixed (Centralized) | 0.235 | 0.248 | 0.254 |
| | FedAvg | 0.231 | 0.248 | 0.264 |
| | FFTS | 0.224 | 0.235 | 0.252 |
| | **FedTRL (Ours)** | **0.216** | **0.230** | **0.241** |
| **Full-shot Forecasting** (MSE on Weather) | All Mixed (Centralized) | 0.227 | 0.235 | 0.238 |
| | FedAvg | 0.228 | 0.239 | 0.240 |
| | FFTS | 0.212 | 0.219 | 0.227 |
| | **FedTRL (Ours)** | **0.204** | **0.211** | **0.214** |
| **Zero-shot Forecasting** (MSE on RW-Bench) | All Mixed (Centralized) | 0.610 | 0.608 | 0.614 |
| | FedAvg | 0.708 | 0.710 | 0.729 |
| | FFTS | 0.605 | 0.614 | 0.614 |
| | **FedTRL (Ours)** | **0.587** | **0.592** | **0.599** |
| **Zero-shot Probabilistic** (MASE on GIFT-eval / FEV) | All Mixed (Centralized) | 0.812 / 0.867 | 0.820 / 0.875 | 0.823 / 0.894 |
| | FedAvg | 0.835 / 0.877 | 0.849 / 0.879 | 0.872 / 0.890 |
| | FFTS | 0.752 / 0.852 | 0.754 / 0.855 | 0.766 / 0.876 |
| | **FedTRL (Ours)** | **0.670 / 0.826** | **0.669 / 0.832** | **0.675 / 0.836** |

