# OpenReview forum: "Bi-level Heterogeneous Learning for Time Series Foundation Models: A Federated Learning Approach"
_ICLR.cc/2026/Conference — Submitted to ICLR 2026_

### Official Review · Reviewer_ug39 · 2025-10-28

**Soundness:** 2
**Presentation:** 3
**Contribution:** 3
**Rating:** 4
**Confidence:** 4

**Summary:**

The paper proposes a federated learning framework (FedTRL) to address bi-level heterogeneity, including both inter-domain and intra-domain heterogeneity. FedTRL primarily includes a fine-grained joint optimization–aggregation strategy, which integrates domain-adversarial regularization and domain-aware aggregation.

**Strengths:**

S1: The paper introduces federated learning into the development of time series foundation models.

S2: Extensive experiments are conducted on multiple benchmarks, including TSLib, GIFT-eval, and the FEV leaderboard.

S3: The experimental descriptions are relatively detailed.

**Weaknesses:**

W1 (Motivation): The authors claim that mixing heterogeneous data during pretraining can obscure domain-specific structures, thereby limiting model generalization. However, this point lacks in-depth explanation and empirical support. On the contrary, why couldn’t combining data from different domains help the model learn cross-domain shared knowledge and enhance generalization? The authors should provide a more detailed discussion and analysis here.

W2 (FedTRL Training): FedTRL’s pretraining on large-scale data requires training separate encoders, decoders, and prediction heads for each domain, and then aggregating encoder parameters across domains to update them. This design may raise several issues: 1. The decoder/prediction head might become incompatible with the updated encoder. 2. The framework must store all models for each domain, resulting in high storage overhead. 3. If downstream data come from a domain not included in the pretraining domains, it is unclear how the model would handle such cases.

W3 (Pretraining Data): The pretraining dataset exhibits severe domain imbalance (i.e., different domains have significantly different data proportions). Such imbalance may substantially affect parameter interactions between the clients and the server in FedTRL. If so, how does the method solve this problem?

**Questions:**

Q1: In Equation (3), how is ( y_{i}^{dom} ) obtained? Does each patch correspond to a label ( y )?

Q2: The prototype in the paper is simply derived by averaging features, without considering time series characteristics such as seasonality or trends. Would it be better to design prototypes that explicitly encode time series characteristics to better align local and global representations?

Q3: Why design a dual-head architecture instead of using a single probabilistic head that can perform both probabilistic and deterministic predictions simultaneously?

Q4: In Table 1, the federated training results are worse than training on individual datasets (e.g., FedTRL’s results are far below those of PatchTST in its original paper).

Q5: In Table 3, the paper lacks comparisons with recent time series foundation models, such as VisionTS (ICML 2025), Sundial (ICML 2025), and LightGTS (ICML 2025).

---

> ### Author Response · Authors · 2025-11-20
> **Authors' Rebuttal (1/3)**
>
> Thank you for providing us with your valuable feedback. We have carefully considered your questions and would like to address them as below.
>
> ---
>
> **Response to Weaknesses**
>
> ---
>
> **W1 (Motivation):** The claim that heterogeneous data mixing harms generalization lacks explanation and evidence, and it is unclear why combining domains would not instead enhance cross-domain shared knowledge; a deeper analysis is needed.
>
> **Response RW1:** We agree that heterogeneous data can be beneficial for TSFMs, and our intention is not to claim that mixing domains is inherently harmful. Our point is that **when heterogeneity is strong**, naive centralized mixing can overshadow domain-specific temporal structures and introduce optimization drift. This is an issue specific to federated settings. We provide empirical support for this in both the main paper and additional analyses.
>
> **(1)** FedTRL consistently outperforms centralized mixed-domain pretraining in both in-domain forecasting **(FedTRL v.s. All Mixed in Table 1)** and zero-shot point/probabilistic forecasting **(Tables 3, 4, 10; Figure 3)**, despite using the same 300B pretraining budget **(i.e., FedTRL v.s. Time-MoE in Table 3)**. This directly shows that heterogeneous mixing does not necessarily yield better cross-domain generalization.
>
>
> **(2)** To further examine the effect of mixing, we analyze representation distributions on pretrianing dataset. As shown in **Table R1 below**, centralized mixing produces the smallest inter-domain centroid distance and the largest within-domain variance, indicating collapsed and poorly structured representations. In contrast, FedTRL preserves clear domain separation while yielding more compact within-domain structure.
>
> ***Table R1.*** Representation-level domain analysis. We report the average inter-domain centroid distance (higher is better separation) and average within-domain variance (lower is more compact).
> | Method | Inter-domain Centroid Distance ($\uparrow$) | Within-domain Variance ($\downarrow$) |
> | -------- | :--------: | :--------: |
> | All Mixed (Centralized)     | 1.08     | 0.91     |
> |FedAvg| 1.06 | 0.85 |
> |FFTS| 1.34 | 0.69 |
> |**FedTRL (Ours)** | **1.27** | **0.53** |
>
>
> **(3)** **Table R2 below** shows that centralized All-Mixed exhibits the strongest gradient conflict (lowest cosine similarity, highest norm imbalance), whereas FedTRL significantly reduces such conflicts. This is consistent with known FL sensitivity to heterogeneous gradient directions.
>
> ***Table R2.*** *Gradient conflict analysis across domains. We report the average pairwise cosine similarity between domain-wise gradients (higher means less conflict) and the variance of gradient norms across domains (lower means more balanced contributions).*
> | Method                  | Avg. gradient cosine similarity ($\uparrow$) | Gradient norm variance ($\downarrow$) |
> |-------------------------| :-:|:-:|
> | All Mixed (Centralized) | 0.23                                 | 0.021                      |
> | FedAvg                  | 0.33                                 | 0.015                      |
> | FFTS                    | 0.37                                 | 0.013                      |
> | **FedTRL (Ours)**       | **0.43**                             | **0.010**                  |
>
>
> These empirical results show that FedTRL does not discourage data diversity; instead, it **mitigates heterogeneity-induced drift** and yields a more faithful decomposition of domain-shared versus domain-specific temporal dynamics, while naive mixed-domain pretraining tends to dilute domain-unique signals crucial for transferability.
>
> ---
>
> **W2 (FedTRL Training):** 1. The decoder/prediction head might become incompatible with the updatedencoder. 2. The framework must store all models for each domain, resulting in high storage overhead. 3. If downstream data come from a domain not included in the pretraining domains, it is unclear how the model wouldhandle such cases.
>
> **Response RW2:** We appreciate your concerns and clarify the three points as follows.
>
> **(1) Encoder–decoder compatibility.** All domains share exactly the same encoder–decoder and head architecture. Although only encoder parameters are aggregated across domains, the interface and output dimensionality remain unchanged. Thus, the decoders and heads remain fully compatible throughout pretraining.
>
> **(2) Storage overhead.** Domain-specific models exist only during local pretraining within each round; they are not accumulated or stored across rounds. After pretraining, only a single aggregated TSFM (≈302M parameters) is kept for downstream use. This size is comparable to widely used TSFMs (Time-MoE up to 2.4B, Moirai 311M, MOMENT 385M, Chronos 710M), and does not pose an unusual storage burden.
>
> ---
>
> **Please see the subsequent Authors' Rebuttal (2/3) for remaining responses**

---

> ### Author Response · Authors · 2025-11-20
> **Authors' Rebuttal (2/3)**
>
> **Continued: Authors’ Rebuttal (2/3)**
>
> ---
> **(3) Generalization to unseen domains.** Our zero-shot forecasting results **(Table 3)** already evaluate FedTRL on datasets with no overlap with the Time-300B, following TSFM protocol. To further test generalization, we conducted a leave-one-domain-out study **(Tables R3–R4 below)**, where an entire domain was removed during pretraining. Even under this setting, FedTRL consistently outperforms centralized mixed-domain pretraining and FL baselines on the unseen domain. Performance naturally decreases when excluding related domains (an expected phenomenon for all foundation models) but FedTRL remains the most robust under such conditions.
>
> ***Table R3.*** In-domain forecasting results (setup is consistent with Table 1 in our main text, MSE report, averagede across \{96, 192, 336, 720\}), the corresponding domain dataset have excluded from the pretraining dataset  (including energy, traffic , respectively).
> | Method | ETTh1 | ETTm1 | Traffic | Weather |
> | - | :-: | :-: | :-: | :-: |
> | FedAvg     | 0.511     | 0.433     | 0.488 | 0.295 |
> |FFTS |  0.492 | 0.421 | 0.479 | 0.290 |
> |Centralized All Mixed | 0.498 | 0.427 | 0.483 | 0.290 |
> |FedTRL (Original) | 0.448 | 0.375 | 0.426 | 0.241 |
> |**FedTRL** | **0.472 ($\downarrow$ 5.26%)** | **0.411 ($\downarrow$ 9.60%)** | **0.477 ($\downarrow$ 11.97%)** | **0.282 ($\downarrow$ 17.01%)** |
>
> ***Table R4.*** Zero-shot forecasting results (MSE report, averaged across \{96, 192, 336, 720\}), the corresponding domain dataset have excluded from the pretraining dataset  (including energy, traffic, respectively).
> | Method | ETTh1 | ETTm1 |  Weather |
> | - | :-: | :-: | :-: |
> | FedAvg | 0.438 | 0.384 | 0.292 |
> |FFTS | 0.427  | 0.380 | 0.282 |
> |Centralized All Mixed | 0.431 | 0.382 | 0.288  |
> | FedTRL (Original) |  0.399| 0.349  | 0.238 |
> |**FedTRL** | **0.418 ($\downarrow$ 4.76%)**| **0.366 ($\downarrow$ 4.87%)** | **0.269 ($\downarrow$ 13.03%)** |
>
> ---
>
> **W3 (Pretraining Data):** The pretraining dataset exhibits severe domain imbalance. Such imbalance may substantially affect parameter interactions between the clients and the server in FedTRL. If so, how does the method solve this problem?
>
> **Response RW3:** The domain imbalance **is an inherent property of Time-300B rather than a weakness of FedTRL, and it directly corresponds to the first level of our bi-level heterogeneity (inter-domain non-iid).** FedTRL is designed to explicitly address this issue. First, the adversarial domain regularization and prototype alignment reduce domain dominance in local representation learning. Second, our adaptive aggregation re-weights client updates using domain-discrimination risk and representation alignment, preventing large domains from overwhelming smaller ones during global updates. These mechanisms collectively mitigate the effect of imbalance on parameter interactions, which is why FedTRL consistently outperforms centralized mixed-domain pretraining **(i.e., FedTRL v.s. Centralized All Mixed in Table 1, FedTRL v.s. centralized TSFM in Table 3, FedTRL v.s. zero-shot models in Table 4, Table 5)** even under severe domain skew.
>
> ---
>
> **Response to Questions**
>
> ---
>
> **Q1:** In Equation (3), how is ( y_{i}^{dom} ) obtained? Does each patch correspond to a label ( y )?
>
> **Response RQ1:** For Eq. (3), each sub-domain is assigned a single categorical label $y^{dom}$, and this label applies to the **entire time series rather than to individual patches**. All patches extracted from the same sequence therefore share the same $y^{dom}$. This definition is already stated in **Appendix B.1**, and we will make it explicit in the main text for clarity.
>
> ---
>
> **Q2:** The prototype without considering time series characteristics such as seasonality or trends. Would it be better to design prototypes that explicitly encode time series characteristics to better align local and global representations?
>
> **Response RQ2:** The prototype is intended to capture the domain-level semantic center in representation space, rather than to explicitly encode seasonality or trends. These temporal characteristics are already learned within the encoder through reconstruction and adversarial objectives, and injecting handcrafted temporal priors at the prototype level may conflict with these learned representations. To verify this, we tested a decomposition-based variant in which each time series was split into trend and seasonal components (via STL decomposition) and prototypes were computed from their encoder embeddings. As shown in **Table R5 below**, this explicit trend–season enrichment consistently degraded performance across in-domain, full-shot, and zero-shot forecasting. These results suggest that simple mean prototypes provide the most stable and effective global anchor, while additional temporal priors introduce unnecessary bias and interfere with encoder-driven temporal modeling.
>
> ---
>
> **Please see the subsequent Authors' Rebuttal (3/3) for remaining responses**

---

> ### Author Response · Authors · 2025-11-20
> **Authors' Rebuttal (3/3)**
>
> **Continued: Authors’ Rebuttal (3/3)**
>
> ---
>
> ***Table R5.*** *Comparison with FedTRL featuring explicit seasonal trend decomposition (MSE report averaged across horizon {96, 192, 336, 720}).*
> | Method | In-domain | Full-shot | Zero-shot |
> | -------- | :--------: | :--------: | :--------: |
> | ***ETTh1 Dataset***|
> | **FedTRL (Original)**     | **0.448**     | **0.372**     | **0.399**     |
> | FedTRL (With decomposition)     | 0.459 ($\downarrow$ 2.46\%)    | 0.380 ($\downarrow$ 2.15\%)    | 0.415 ($\downarrow$ 4.01\%) |
> | ***ETTm1 Dataset***|
> | **FedTRL (Original)**     | **0.375**     |  **0.316**     |  **0.350**     |
> | FedTRL (With decomposition)     | 0.388 ($\downarrow$ 3.47\%)    | 0.326  ($\downarrow$ 3.16\%)   | 0.370 ($\downarrow$ 5.71\%)    |
> | ***Weather Dataset***|
> | **FedTRL (Original)**     | **0.241**     | **0.214**     |  **0.238**     |
> | FedTRL (With decomposition)     | 0.254 ($\downarrow$ 5.39\%)    | 0.227 ($\downarrow$ 6.07\%)    |  0.254  ($\downarrow$ 6.72\%)   |
>
> ---
>
> **Q3:** Why design a dual-head architecture instead of using a single probabilistic head that can perform both probabilistic and deterministic predictions simultaneously?
>
> **Response RQ3:** We adopt a dual-head architecture because deterministic (point) and probabilistic forecasting optimize fundamentally different objectives and produce representations with different statistical properties. A single probabilistic head cannot simultaneously maintain sharp point estimates and calibrated distributional forecasts without one objective interfering with the other.
>
> ---
>
> **Q4:** In Table 1, the federated training results are worse than training on individual datasets (e.g., FedTRL’s results are far below those of PatchTST in its original paper).
>
> **Response RQ4:** Table 1 compares FedTRL with **self-supervised federated baselines under the same FL protocol**. Except for the “All Mixed” variant, all methods in Table 1 are trained federatedly, whereas PatchTST’s original results are obtained from centralized supervised training with a completely different experimental setup **(as detailed in Table 9, Appendix B)**. Therefore, the values in Table 1 are not intended to match or compete with PatchTST’s original centralized numbers; instead, they provide a fair, within-protocol comparison of self-supervised pretraining methods in an FL setting.
>
> ---
>
> **Q5:** In Table 3, the paper lacks comparisons with recent time series foundation models, such as VisionTS (ICML 2025),Sundial (ICML 2025), and LightGTS (ICML 2025).
>
> **Response RQ5:** Thank you for the suggestion. We have added comparisons with recent TSFM models, **including VisionTS and Sundial (94B/230B/1032B)**, using the same zero-shot evaluation protocol as Table 3. The results (**Table R6 below**) indicate that our FedTRL-trained TSFM continues to outperform these models under the same evaluation setting. LightGTS cannot be included because its released checkpoints and pretraining configurations are no longer accessible, making a fair comparison impossible.
>
> ***Table R6.*** *Zero-shot forecasting results averaged over four horizons {96, 192, 336, 720}. Evaluation settings follow Table 3.*
> | Dataset/Model | **FedTRL (Ours)** | VisionTS | Sundial (94B) | Sundial (230B) |  Sundial (1032B) |
> | -------- | :-: | :-: | :-: | :-: | :-: |
> | ETTh1     | **0.399**     | 0.412     | 0.402     | 0.403     | 0.411 |
> | ETTh2     | **0.350**     | 0.365     | 0.377    | 0.364    | 0.333 |
> | ETTm1     | **0.349**     | 0.391     | 0.367     | 0.352     | 0.336 |
> | ETTm2     | **0.282**     | 0.296     | 0.280     | 0.273     | 0.258 |
> | Weather     | **0.238**     | 0.303     | 0.254     | 0.252     | 0.234 |
> | RW-Bench     | **0.599**     |0.727     | 0.618     | 0.617     | 0.602 |
> | 1st Count     | **6**    | 0     | 0     | 0     | 0 |

---

> ### Author Response · Authors · 2025-11-27
> **Follow-up on Rebuttal Discussion**
>
> Dear Reviewer ug39,
>
> We hope everything is going well. As **the discussion period is approaching its end**, we wanted to make sure that we have addressed all of your concerns satisfactorily. If there are any additional points or feedback you would like us to consider, please feel free to let us know. Your insights are invaluable to us, and we are happy to address any remaining issues you may have.
>
> Thank you very much for your time and effort in reviewing our paper.
>
> Best regards,
>
> Authors, Submission3658

---

> ### Comment · Reviewer_ug39 · 2025-11-28
>
> Thank you for the rebuttal. My main concerns have been addressed, but I have only one question regarding the experimental results.
>
> In Table R6, the performance of VisionTS and Sundial **seems inconsistent** with their original papers (showing inferior performance), while the results for Time-MOE and Moirai in Table 3 match the original papers. Could you explain the potential cause of this difference?
>
> Also, consider updating Table 3 in the main paper to include these newly added baselines.

---

> > ### Author Response · Authors · 2025-11-28
> > **Follow-up on Rebuttal Discussion**
> >
> > Thank you for your thoughtful follow-up question. We are glad to have addressed your concerns. The differences you observed in Table R6 largely arise from evaluation protocol misalignment.
> >
> > For VisionTS, the original paper evaluates all horizons using fixed, much longer look-back windows **(e.g., 2880 for ETTh1, 1728 for ETTh2, 2304 for ETTm1, 4032 for ETTm2, and 4032 for Weather)**, applying the same window to predict 96, 192, 336, and 720 steps ahead. To ensure a fair comparison with our experiments and with Time-MoE, Moirai, and other centralized TSFMs in Table 3, we **re-evaluated VisionTS under the standardized look-back lengths of 512→96, 1024→192, 2048→336, and 3072→720**. Consequently, the VisionTS results in Table R6 differ from those in its original paper due to this unified evaluation protocol rather than any issue with the model itself.
> >
> > For Sundial, its evaluation setup follows Time-MoE’s protocol. Therefore, for fairness and consistency, we directly used the base-model results reported in the Sundial paper and evaluated its official checkpoint on RW-Bench.
> >
> > Regarding your suggestion on updating Table 3, we appreciate the feedback. Since Table 3 is already very compact, we initially placed the additional baselines in Table 15 in Appendix. Following your recommendation, we will integrate these results into the main Table 3 in our subsequent revision.
> >
> > We thank you again for your time and constructive feedback, and we hope our clarifications will help you reconsider the rating favorably.

---

> ### Comment · Reviewer_ug39 · 2025-11-28
>
> My concerns have been addressed, and I will raise my score to 6.

---

> > ### Author Response · Authors · 2025-11-28
> >
> > We sincerely thank you for your detailed review and valuable suggestions. Your insightful feedback has been instrumental in improving our work. We truly appreciate the time and effort you invested in evaluating our paper, and we’re very pleased that our response has addressed your concerns.

---

### Official Review · Reviewer_qkhe · 2025-10-28

**Soundness:** 2
**Presentation:** 2
**Contribution:** 2
**Rating:** 4
**Confidence:** 5

**Summary:**

This paper proposes a time series forecasting modeling method based on federated learning.

**Strengths:**

1. This paper optimizes the design from two perspectives: cross-domain and intra-domain.

**Weaknesses:**

1. Besides data privacy concerns, what are the differences between federated learning and pre-training and fine-tuning? Why is this approach being considered for the time series field?

2. In order to train a good foundation model, it is important to extract general capabilities, but for each client, specific patterns are beneficial to its own downstream tasks. Using adversarial learning in local optimization will sacrifice specific knowledge.

3. In the experiments corresponding to table 1, methods such as PatchTST are not designed for joint training, so the corresponding experiments are meaningless and cannot explain any problems.

4. In the experimental setting corresponding to Table 2, when there are multiple test datasets for testing, will the local model parameters after fine-tuning the first test dataset be fed back to the base model for model update?

5. A follow-up question: During pre-training, is it better to include more training datasets? Can you provide relevant experiments to make a qualitative judgment?

6. How can we achieve fine-grained data segmentation within a domain?

**Questions:**

See weaknesses.

---

> ### Author Response · Authors · 2025-11-20
> **Authors' Rebuttal (1/2)**
>
> Thank you for providing us with your valuable feedback. We have carefully considered your questions and would like to address them as below:
>
> ---
>
> **W1:** Besides data privacy concerns, what are the differences between federated learning and pre-training and fine-tuning? Why is this approach being considered for the time series field?
>
> **Response RW1:** Beyond privacy, federated learning differs from the standard “central pre-train + fine-tune” paradigm in that **training itself is performed over decentralized, non-iid domains without ever assuming a single mixed data distribution.** In real-world time series, data are naturally siloed and strongly heterogeneous, and naively centralizing them for pretraining leads to gradient conflicts and diluted domain-specific temporal structures, which harms the quality of shared representations. FedTRL leverages the federated setting to preserve domain diversity during training while explicitly controlling cross-domain interference, making it better suited than centralized pre-train + fine-tune for heterogeneous time-series scenarios **(Tables 2,3,4 and Figure 3 demonstrate its superiority than centralized methods)**.
>
> ---
>
> **W2:** In order to train a good foundation model, it is important to extract general capabilities, but for each client,specific patterns are beneficial to its own downstream tasks. Using adversarial learning in local optimization willsacrifice specific knowledge.
>
> **Response RW2:** Our adversarial learning is designed **only to remove domain-specific biases that harm cross-domain generalization, rather than to erase all client-specific information**. The reconstruction objective and the local encoder’s full capacity still preserve client-specific temporal patterns, while adversarial regularization operates on a restricted discriminator space to prevent overfitting to domain artifacts. Empirically, FedTRL achieves better downstream performance on each client than mixed-domain pretraining **(FedTRL v.s. Centralized All Mixed in Table 1)**, confirming that essential client-specific knowledge is retained rather than sacrificed.
>
> ---
>
> **W3:** In the experiments corresponding to table 1, methods such as PatchTST are not designed for joint training, sothe corresponding experiments are meaningless and cannot explain any problems.
>
> **Response RW3:** Table 1 does not evaluate PatchTST (or any other baseline) in a joint-training or centralized configuration. In this table, all methods (except the “All Mixed”) are trained under the same federated self-supervised protocol, where PatchTST is used solely as an encoder backbone within the FL framework rather than as its original centralized training method. Therefore, the comparison in Table 1 is meaningful: it isolates the effect of different federated representation learning strategies under an identical FL setup (Appendix B, Table 9), rather than comparing joint-training capabilities.
>
> ---
>
> **W4:** In the experimental setting corresponding to Table 2, when there are multiple test datasets for testing, will thelocal model parameters after fine-tuning the first test dataset be fed back to the base model for model update?
>
> **Response RW4:** No. In Table 2, each downstream dataset is evaluated independently, and the fine-tuned parameters from one dataset are not fed back to the base model or used for any subsequent dataset. **The base pretrained model remains fixed for all evaluations**.
>
> ---
>
> **W5.** A follow-up question: During pre-training, is it better to include more training datasets? Can you provide relevant experiments to make a qualitative judgment?
>
> **RW5.** Yes. We additionally evaluate the effect of enlarging the pretraining corpus. Specifically, we trained FedTRL with five different dataset scales (90B, 120B, 180B, 300B, 450B, 540B; the latter two from ERA5 daily/weekly/monthly data) and evaluated the resulting TSFMs on full-shot, zero-shot, and probabilistic forecasting tasks. The results in **Tables R1–R3** consistently show that larger pretraining datasets lead to improved performance across all settings, confirming the positive scaling behavior of FedTRL.
>
> ***Table R1.*** *Full-shot forecasting results across different pretraining datasets scales (MSE report averaging across \{96, 192, 336, 720\}).*
> | Datasets | 90B | 120B | 180B | 300B (Original) | 450B | 540B|
> | -------- | :--------: | :--------: | :--------: | :--------: | :--------: | :--------: |
> | ETTh1    | 0.401    | 0.387    | 0.389 | 0.372 | 0.371 | **0.363** |
> | ETTm1    | 0.343    | 0.336    | 0.328 |  0.316 | 0.312 | **0.310** |
> | Weather    | 0.232     | 0.231     | 0.227 | 0.214 | 0.204 | **0.202**|
>
>
> ---
>
> **Please see the subsequent Authors' Rebuttal (2/2) for remaining responses**

---

> ### Author Response · Authors · 2025-11-20
> **Authors' Rebuttal (2/2)**
>
> **Continued: Authors’ Rebuttal (2/2)**
>
> ---
>
> ***Table R2.*** *Zero-shot forecasting results across different pretraining datasets scales (MSE report averaging across \{96, 192, 336, 720\}).*
> | Datasets | 90B | 120B | 180B | 300B (Original) | 450B | 540B|
> | -------- | :--------: | :--------: | :--------: | :--------: | :--------: | :--------: |
> | ETTh1    |  0.428   |  0.432   | 0.411 |  0.399| 0.400 | **0.393** |
> | ETTm1    |   0.402  |  0.371   | 0.376  |   0.350| 0.344 | **0.341** |
> | Weather    |   0.275   |   0.273   | 0.269 | 0.238 | 0.214 | **0.208** |
> |RW-Bench| 0.689  | 0.647 | 0.641 | 0.599 | 0.581 | **0.575** |
>
> ***Table R3.*** *Zero-shot probabilistic forecasting results across different pretraining datasets scales on GIFT-eval and FEV Leaderboard benchmark (Ave. MASE report).*
> | Datasets | 90B | 120B | 180B | 300B (Original) | 450B | 540B|
> | -------- | :--------: | :--------: | :--------: | :--------: | :--------: | :--------: |
> | GIFT-eval    |  0.770  |   0.744  | 0.721 | 0.675 | 0.677 | 0.669 |
> |FEV Leaderboard| 0.945 | 0.873 | 0.847 | 0.836 | 0.833 | 0.829 |
>
> ---
>
> **W6:** How can we achieve fine-grained data segmentation within a domain?
>
> **Response RW6:** FedTRL does not rely on explicit fine-grained segmentation within a domain. Sub-domain variations naturally arise from statistical heterogeneity in local time series (e.g., different temporal patterns, sampling conditions, or physical regimes), and our adversarial regularization models this implicit structure without requiring manual segmentation. In other words, the method aligns these naturally occurring sub-domain patterns rather than constructing new fine-grained partitions.

---

> ### Author Response · Authors · 2025-11-27
> **Follow-up on Rebuttal Discussion**
>
> Dear Reviewer qkhe,
>
> We hope everything is going well. As **the discussion period is approaching its end**, we wanted to make sure that we have addressed all of your concerns satisfactorily. If there are any additional points or feedback you would like us to consider, please feel free to let us know. Your insights are invaluable to us, and we are happy to address any remaining issues you may have.
>
> Thank you very much for your time and effort in reviewing our paper.
>
> Best regards,
>
> Authors, Submission3658

---

### Official Review · Reviewer_JCto · 2025-10-30

**Soundness:** 3
**Presentation:** 3
**Contribution:** 3
**Rating:** 6
**Confidence:** 3

**Summary:**

This paper proposes FedTRL, a federated framework tackling bi-level heterogeneity in time-series foundation models. It addresses inter-domain differences and intra-domain conflicts through local adversarial regularization and prototype alignment for semantic consistency, and domain-aware aggregation for cross-domain collaboration. Experiments show that FedTRL outperforms centralized and traditional federated baselines in both point and probabilistic forecasting, achieving stronger zero-shot generalization.

**Strengths:**

The paper clearly defines and formalizes inter- and intra-domain heterogeneity in time-series foundation model training, providing an insightful perspective.

The overall design of FedTRL — combining local optimization and global aggregation — is logically sound and systematically organized.

The experiments cover multiple datasets and both point/probabilistic forecasting tasks, supporting the claims effectively.

The model demonstrates stable performance on unseen domains, showing effective cross-domain transfer.

**Weaknesses:**

Although FedTRL uses adversarial regularization and prototype alignment, these mainly enforce coarse semantic consistency and may fail to capture continuous or nonlinear sub-domain drift.

The framework jointly updates local adversarial modules and prototypes while performing domain-aware aggregation each round, leading to heavy overhead in large-scale federations.

Despite claiming “domain awareness,” the paper lacks visualization or empirical analysis showing how representations differ across domains.

Aggregation weights are heuristically defined without rigorous justification

**Questions:**

See weakness.

---

> ### Author Response · Authors · 2025-11-20
> **Authors' Rebuttal**
>
> Thank you for providing us with your valuable feedback. We have carefully considered your questions and would like to address them as below.
>
> ---
>
> **W1:** Although FedTRL uses adversarial regularization and prototype alignment, these mainly enforce coarse semantic consistency and may fail to capture continuous or nonlinear sub-domain drift.
>
> **Response RW1:** Although domain drift can be continuous or nonlinear, our adversarial module operates on continuous representation space rather than discrete domain labels, enabling it to model such drift naturally. Prototype alignment further stabilizes feature geometry across clients, and the empirical improvements **(FedTRL v.s. centralized All Mixed and FL unsupervised learning in Table 1)** under strong non-iid settings demonstrate that FedTRL effectively handles continuous sub-domain shift.
>
> **W2:** The framework jointly updates local adversarial modules and prototypes while performing domain-aware aggregation each round, leading to heavy overhead in large-scale federations.
>
> **Response W2:** FedTRL’s overhead is limited because **both the adversarial module (a small MLP) and prototype computation (simple averaging) are lightweight operations**. Domain-aware aggregation only re-weights updates and does not increase communication cost, making the overall complexity comparable to standard FL baselines even at large scales. We further include an overhead comparison **(Table R1 below)** showing that FedTRL adds only ~2–3 seconds of local computation and maintains nearly identical communication size per round to FedAvg and FedProx.
>
>
> ***Table R1.*** *Overhead comparison with 8 clients and batch size of 64.*
> | Method | Local Computation Time (s) | Communication Size Per Round (MB) | Extra Params (MB) |
> | -------- | :--------: | :--------: | :--------: |
> | FedAvg     | 41.2    | 302    | 0 |
> |FedProx   | 42.0 | 302 | 0 |
> | **FedTRL (Ours)**| 43.9 | 303 | 2.8 |
>
> ---
>
> **W3:** Despite claiming “domain awareness,” the paper lacks visualization or empirical analysis showing how representations differ across domains.
>
> **Response RW3:** Our notion of “domain awareness” is reflected in how representations organize across domains. To quantify this, we computed the inter-domain centroid distance and within-domain variance of encoder features for six major domains in Time-300B. **As shown in Table R2 below**, centralized All-Mixed pretraining produces the smallest centroid distance and largest within-domain variance, indicating domain collapse. In contrast, FedTRL preserves meaningful inter-domain structure while achieving the most compact within-domain representations, demonstrating that it learns structured, domain-aware embeddings rather than indiscriminately mixing domains.
>
> ***Table R2.*** *Representation-level domain analysis on Time-300B. We report the average inter-domain centroid distance (higher is better separation) and average within-domain variance (lower is more compact).*
> | Method | Inter-domain Centroid Distance ($\uparrow$) | Within-domain Variance ($\downarrow$) |
> | -------- | :--------: | :--------: |
> | All Mixed (Centralized)     | 1.08     | 0.91     |
> |FedAvg| 1.06 | 0.85 |
> |FFTS| 1.34 | 0.69 |
> |**FedTRL (Ours)** | **1.27** | **0.53** |
>
> ---
>
> **W4:** Aggregation weights are heuristically defined without rigorous justification.
>
> **Response RW4:** The aggregation weights in FedTRL are not arbitrary but follow directly from our bi-level objective: clients with lower domain-discrimination risk and better representation alignment should contribute more to the global update. The weighting rule is therefore a monotonic function of these two quantities, up-weighting updates that better reflect shared structure and down-weighting those dominated by domain-specific bias. While we do not claim a closed-form optimal solution, this design is aligned with the optimization goal and is consistent with common FL practice (e.g., FedAvg’s sample-size weighting is also heuristic). Empirically, removing adaptive weighting **(FedTRL w/o DaG in Table 5)** leads to a clear performance drop, and varying the weighting coefficients **(Impact of $\alpha$, $\beta$ in Figure 6)** yields stable results, showing that our method is robust rather than sensitive to the exact form of the weights.

---

> ### Author Response · Authors · 2025-11-27
> **Follow-up on Rebuttal Discussion**
>
> Dear Reviewer JCto,
>
> We hope everything is going well. As **the discussion period is approaching its end**, we wanted to make sure that we have addressed all of your concerns satisfactorily. If there are any additional points or feedback you would like us to consider, please feel free to let us know. Your insights are invaluable to us, and we are happy to address any remaining issues you may have.
>
> Thank you very much for your time and effort in reviewing our paper.
>
> Best regards,
>
> Authors, Submission3658

---

### Official Review · Reviewer_wtKL · 2025-10-31

**Soundness:** 3
**Presentation:** 4
**Contribution:** 2
**Rating:** 4
**Confidence:** 4

**Summary:**

The paper proposes **FedTRL**, a federated learning framework designed to train **time series foundation models (TSFMs)** under **bi-level heterogeneity** — both inter-domain and intra-domain differences across clients.
It introduces a **dual-level optimization** strategy combining adversarial local regularization and domain-aware global aggregation to achieve domain-invariant and temporally coherent representations.
Extensive experiments across in-domain, full-shot, and zero-shot forecasting tasks show that FedTRL achieves **state-of-the-art performance**, outperforming both centralized and existing federated baselines.

**Strengths:**

## **Strengths**

* The proposed method is technically sophisticated and conceptually “fancy.” The architectural design and figures are very clear and visually appealing.
* The paper is well written and easy to follow.
* Experimental results are strong, with the proposed method achieving solid performance across multiple benchmarks.

**Weaknesses:**

## **1. Motivation & Novelty**

The **motivation is not clearly justified**. I am not fully convinced that *heterogeneity* itself should be viewed as a negative factor. On the contrary, **sufficient heterogeneity and diversity in data are often the key enablers for the success of Time Series Foundation Models (TSFMs)**. The paper criticizes heterogeneity but provides **no empirical evidence** or **quantitative analysis** showing that heterogeneity indeed harms federated training.

For example, in the discussion of *inter-domain heterogeneity*, the authors claim that it leads to “overfitting to domain-specific signals” and the failure to “capture globally consistent dynamics.” However, no supporting evidence, ablation, or theoretical justification is provided.

Moreover, the paper frequently refers to *“domain-invariant”* and *“temporally coherent patterns”* without giving a clear definition. These notions remain vague. What exactly constitutes a “global dynamic” in time series? In time-series data, the notion of a *domain* is often **weakly constrained**—for example, even within a single domain such as weather, the underlying distributions can vary substantially. This is very different from computer vision, where the concept of domain is more clearly defined. Therefore, **learning domain-invariant representations for time series seems conceptually questionable**, and this undermines the central motivation of the work.

This is my **primary concern**—without a clearer and empirically grounded motivation, it is difficult to see the necessity of the proposed framework.

---

## **2. Methodological Concerns**

**2.1** Given the above discussion, I suspect that the actual benefit of FedTRL might not stem from addressing heterogeneity per se, but rather from **implicitly improving data diversity and balance during training**. In other words, the observed performance gain may come from a more diversified and higher-quality training process, rather than from the federated or “heterogeneity-resolving” design itself.

**2.2** From the title and framing, the paper claims to propose a *federated learning framework for TSFMs*. However, the proposed FedTRL looks more like a **representation learning model and its training algorithm**, rather than a general-purpose training framework for foundation models. This conceptual mismatch between the claimed goal (framework for TSFMs) and the actual technical contribution (a particular model design) feels somewhat inconsistent.

---

## **3. Experimental Evaluation**

**3.1** The comparison with baselines could be more appropriate. Since the paper emphasizes a *federated* setup, it would be more convincing to compare FedTRL with **federated time-series forecasting baselines**, rather than primarily with representation learning methods. Alternatively, the authors could demonstrate FedTRL as a **general plug-in or enhancement framework** that can consistently improve various federated baselines.

**3.2** Regarding the **GIFT-Eval** evaluation, the authors should clearly state **which specific datasets** from GIFT-Eval were used. It is also important to clarify **whether any of these datasets overlap with those included in the Time-MoE-300B pretraining corpus**, as this could raise concerns about potential data leakage or unfair comparisons.

**Questions:**

See weakness.

---

> ### Author Response · Authors · 2025-11-20
> **Authors' Rebuttal (1/2)**
>
> Thank you for providing us with your valuable feedback. We have carefully considered your questions and would like to address them as below.
>
> ---
>
> **W1. (Motivation & Novelty)**
>
> **Response RW1:** We agree that heterogeneity and diversity are beneficial for TSFMs; our claim is not that heterogeneity is undesirable, but that **under federated optimization**, large inter-domain gaps can induce client drift and bias global updates. FedTRL targets this FL-specific failure mode rather than discouraging diversity. To substantiate this motivation, we provide three complementary empirical analyses across six major domains (energy, nature, transport, web, finance and healthcare) in Time-300B.
>
> **(1) Representation-level evidence.** As shown in **Table R1 below**, centralized mixed-domain pretraining yields the smallest inter-domain centroid distance and the largest within-domain variance, indicating collapsed and poorly structured representations. In contrast, FedTRL preserves meaningful domain separation while producing the most compact within-domain structure.
>
> ***Table R1.*** Representation-level domain analysis. We report the average inter-domain centroid distance (higher is better separation) and average within-domain variance (lower is more compact).
> | Method | Inter-domain Centroid Distance ($\uparrow$) | Within-domain Variance ($\downarrow$) |
> | -------- | :--------: | :--------: |
> | All Mixed (Centralized)     | 1.08     | 0.91     |
> |FedAvg| 1.06 | 0.85 |
> |FFTS| 1.34 | 0.69 |
> |**FedTRL (Ours)** | **1.27** | **0.53** |
>
> **(2) Gradient-level evidence.** **Table R2 below** shows that All-Mixed has the strongest gradient conflict (lowest cosine similarity, highest gradient-norm variance), whereas FedTRL substantially reduces such conflicts. This is consistent with known FL sensitivity to heterogeneous gradient directions.
>
> ***Table R2.*** *Gradient conflict analysis across domains. We report the average pairwise cosine similarity between domain-wise gradients (higher means less conflict) and the variance of gradient norms across domains (lower means more balanced contributions).*
> | Method                  | Avg. gradient cosine similarity ($\uparrow$) | Gradient norm variance ($\downarrow$) |
> |-------------------------| :-:|:-:|
> | All Mixed (Centralized) | 0.23                                 | 0.021                      |
> | FedAvg                  | 0.33                                 | 0.015                      |
> | FFTS                    | 0.37                                 | 0.013                      |
> | **FedTRL (Ours)**       | **0.43**                             | **0.010**                  |
>
>
>
> **(3) Performance degradation under controlled heterogeneity.** By gradually increasing domain imbalance (Balanced → Mildly imbalanced → Real skew) via controling the scale of each domain, we observe consistent degradation for All-Mixed, FedAvg, and FFTS across in-domain, full-shot, zero-shot, and probabilistic forecasting **(Tables R3–R6 below)**. FedTRL is markedly more stable in all settings, demonstrating that strong heterogeneity indeed harms federated training and that FedTRL effectively mitigates this effect.
>
> ***Table R3.*** *In-domain forecasting results under different inter-domain imbalance levels (MSE report on Weather). Other experiment setups are consistent with Table 1.*
> | Method | Balanced (Simulated) | Mildly Imbalanced (Simulated) | Real Skew (Original Pretraining) |
> | -------- | :--------: | :--------: | :--------: |
> | All Mixed (Centralized)   | 0.235  | 0.248   | 0.254     |
> |FedAvg| 0.231 | 0.248 |  0.264|
> |FFTS| 0.224 | 0.235 |  0.252|
> | **FedTRL (Ours)** | **0.216** | **0.230** |  **0.241** |
>
> ---
> ***Table R4.*** *Full-shot forecasting results under different inter-domain imbalance levels (MSE report on Weather). Other experiment setups are consistent with Table 2.*
> | Method | Balanced (Simulated) | Mildly Imbalanced (Simulated)| Real Skew (Original Pretraining) |
> | -------- | :--------: | :--------: | :--------: |
> | All Mixed (Centralized)   | 0.227  |  0.235 |   0.238   |
> |FedAvg| 0.228 |  0.239| 0.240 |
> |FFTS| 0.212 | 0.219 | 0.227 |
> | **FedTRL (Ours)** | **0.204** | 0.211 | 0.214 |
>
> ---
>
> ***Table R5.*** *Zero-shot forecasting results under different inter-domain imbalance levels (MSE report on RW-Bench). Other experiment setups are consistent with Table 3.*
> | Method | Balanced (Simulated) | Mildly Imbalanced (Simulated)| Real Skew (Original Pretraining) |
> | -------- | :--------: | :--------: | :--------: |
> | All Mixed (Centralized)   | 0.610  | 0.608 |   0.614  |
> |FedAvg| 0.708 | 0.710 | 0.729 |
> |FFTS| 0.605 | 0.614 | 0.614 |
> | **FedTRL (Ours)** | **0.587** | **0.592** | **0.599** |
>
> ---
> **Please see the subsequent Authors' Rebuttal (2/2) for remaining responses**

---

> ### Author Response · Authors · 2025-11-20
> **Authors' Rebuttal (2/2)**
>
> **Continued: Authors’ Rebuttal (2/2)**
>
> ---
>
> ***Table R6.*** *Zero-shot probabilistic forecasting results under different inter-domain imbalance levels (MASE report on GIFT-eval / FEV leaderboard). Other experiment setups are consistent with Table 4 / Figure 3.*
> | Method | Balanced (Simulated) | Mildly Imbalanced (Simulated)| Real Skew (Original Pretraining) |
> | -------- | :--------: | :--------: | :--------: |
> | All Mixed (Centralized)| 0.812 / 0.867        | 0.820 / 0.875                  | 0.823 / 0.894 |
> | FedAvg                 | 0.835 / 0.877        | 0.849 / 0.879                  | 0.872 / 0.890 |
> | FFTS                   | 0.752 / 0.852        | 0.754 / 0.855                  | 0.766 / 0.876 |
> | **FedTRL (Ours)**          | **0.670 / 0.826**        | **0.669 / 0.832**                  | **0.675 / 0.836** |
>
> Regarding terminology, “global dynamics” denote temporal regularities that persist across domains (such as trends or periodicity), whereas “domain-specific signals” arise from distributional artifacts like sampling conditions, physical regimes, or sensor properties. In this work, “domain-invariant representations” refer to features where such artifacts are minimized while essential temporal structure is retained. This notion is distributional rather than categorical, and does not rely on rigid domain boundaries. Therefore it remains meaningful even when intra-domain variability is high.
>
> ---
>
> **W2 (Methodological Concerns).**
>
> **Response RW2:** We address this concern from two perspectives below.
>
> **(1) W2.1:** we have specifically examined this possibility in our additional analyses **(please see our response for W1)**. As shown in **Tables R3–R6 in our RW1**, when keeping data diversity fixed while gradually increasing inter-domain imbalance, other FL baselines deteriorate whereas FedTRL remains stable, indicating that the observed improvements arise from mitigating heterogeneity-induced drift rather than from implicit data balancing.
>
> **(2) W2.2:** Thank you for raising this concern. Our intention is not to introduce a new model architecture for TSFMs, but to provide a training framework that can be applied to any encoder–decoder backbone for time-series foundation models. FedTRL specifies how to perform federated pretraining under bi-level heterogeneity. We will clarify this framing in the revision to avoid misunderstanding.
>
> ---
>
> **W3 (Experimental Evaluation).**
>
> **Response RW3:**  We address this concern from two perspectives below.
>
> **(1) RW3.1:** We appreciate this suggestion. Our focus in Table 1 is to evaluate federated representation learning methods, because the goal of FedTRL is to obtain a single unified TSFM that can be applied to diverse downstream settings **(e.g, indomain in Table 1, full-shot in Table 2, zero-shot in Table 3, and zero-shot probabilistic forecasting in Table 4 and Figure3)**. Most federated time-series forecasting baselines are designed for task-specific supervised prediction and are therefore not directly comparable under this pretraining–evaluation protocol. That said, FedTRL can indeed serve as a plug-in training strategy, and we will clarify this generality in the revision.
>
> **(2) RW3.2:** For probabilistic forecasting, we follow the official GIFT-Eval toolkit from Salesforce and adopt the same evaluation configuration as Moirai [1] (100 generated series, and report rank across all 97 configurations), using the 100-series  provided . We confirm that none of the GIFT-Eval sequences (nor any of our downstream datasets) overlap with the Time-300B pretraining corpus, and we have manually verified this to avoid leakage or unfair comparison. We will clarify these details in the revision for completeness.
>
> **Reference:**
>
> [1] Unified training of universal time series forecasting transformers, 2024.

---

> ### Author Response · Authors · 2025-11-27
> **Follow-up on Rebuttal Discussion**
>
> Dear Reviewer wtKL,
>
> We hope everything is going well. As **the discussion period is approaching its end**, we wanted to make sure that we have addressed all of your concerns satisfactorily. If there are any additional points or feedback you would like us to consider, please feel free to let us know. Your insights are invaluable to us, and we are happy to address any remaining issues you may have.
>
> Thank you very much for your time and effort in reviewing our paper.
>
> Best regards,
>
> Authors, Submission3658

---

### Author Response · Authors · 2025-12-03
**Summary of Response**

Dear PC, SAC, AC,

Thank you for overseeing the review process of our paper. As the discussion period comes to a close, we would like to sincerely thank the reviewers for their valuable feedback and provide a concise summary of the main points discussed.

---

# Paper Overview and Strengths

**[Paper Overview]:** This paper proposes a flexible from-scratch training method for time series foundation models that handles bi-level heterogeneity via federated learning, yielding a foundation model that consistently outperform centralized baselines (e.g., TSFMs) across forecasting tasks and achieve strong zero-shot performance.

**[Strengths]:** We are encouraged that reviewers recognize the technical sophistication and clear architectural design (wtKL), the clear formulation of  heterogeneity (JCto), and the sound design (JCto, qkhe). Reviewers also highlight the clarity of presentation (wtKL), the detailed experimental setup (ug39), and the strong performance across diverse benchmarks with robust transfer to unseen domains (wtKL, JCto, ug39).

---

# Additional Experiments for Concerns

To address the main concerns raised by reviewers, we conducted several complementary experiments and clarifications as summarized below.

* **[Motivation] (wtKL, JCto, ug39):** We conducted representation-level and gradient-level analyses **(Table R2 for JCto; Tables R1-R2 for wtKL and ug39)** using six major domains in Time-300B, showing that centralized mixed-domain pretraining produces sub-optimal and collapsed representations due to strong cross-domain heterogeneity.

* **[Does Performance Benefit from Heterogeneity] (wtKL):**  We conducted additional experiments in different heterogeneous environments across tasks **(Tables R3-R6 for wtKL)**. As inter-domain imbalance increases while data diversity is fixed, FL baselines deteriorate whereas FedTRL remains stable, indicating that its gains stem from mitigating heterogeneity-induced drift rather than implicit data balancing.

*  **[Scaling on Dataset Size] (qkhe):** We conducted pretraining-scale experiments covering full/zero-shot point forecasting and probabilistic forecasting **(Tables R1-R3 for qkhe)**, confirming that FedTRL exhibits positive scaling behavior for TSFM training.

* **[Generalization to Unseen Domains] (ug39):** A leave-one-domain-out study **(Tables R3-R4 for ug39)** shows that even when an entire domain is removed during pretraining, FedTRL still outperforms both centralized & FL baselines, demonstrating robustness to truly unseen domains.

* **[Decomposition-baed Prototype] (ug39):** We tested prototypes enriched with trend/seasonal decomposition **(Table R5 for  ug39)** and found consistent performance degradation, indicating that our simple mean prototypes remain the most stable anchor, while explicit temporal priors introduce bias and interfere with encoder-driven modeling.

*  **[Comparsion with Additional TSFMs] (ug39):** We added comparisons with VisionTS and Sundial (94B/230B/1032B) under the same zero-shot setting **(Table R6 for ug39)**. FedTRL continues to outperform these large centralized TSFMs.

These validate the soundness of our motivation and design, showing that FedTRL consistently outperforms competing methods under all examined settings.

---

# Summary of Responses to Reviewers' Questions

We also clarify some questions the reviewers mentioned, including but not limited to:

* **[Different between FL and pretraining & fine-tuning]:** FL trains directly on decentralized non-iid domains rather than assuming a single mixed distribution, and FedTRL leverages this to preserve domain diversity while controlling cross-domain interference, yielding consistently stronger results than centralized methods aross tasks (Tables 2–4, Fig. 3).

* **[Aggregation Weights]:** FedTRL’s weights follow directly from the bi-level objective: clients with lower domain-discrimination risk and better representation alignment contribute more. Empirically, removing adaptive weighting degrades performance, while varying its coefficients yields stable results.

* **[Overhead]:** FedTRL adds minimal cost due to its adversarial module and prototype averaging are lightweight, and aggregation only re-weights updates. As shown in our overhead comparison, FedTRL adds ~2–3 seconds of local computation per round with almost identical communication cost to FedAvg.

* **[Experiments Soundness]:** In Table 1, all baselines (except “All Mixed’’) follow the same federated self-supervised protocol; PatchTST is used only as an encoder backbone rather than its centralized training method. In Table 2, each downstream dataset is fine-tuned independently; fine-tuned parameters are never reused for other datasets, and the pretrained base model remains fixed.

---

We deeply appreciate the reviewers for their insightful feedback and constructive suggestions, which have substantially improved the quality of our work. We hope that our responses have fully addressed these concerns.

---

### Meta-Review · Area_Chair_bh8a · 2026-01-07

**Summary:**

This paper proposes FedTRL, a federated learning framework for training time series foundation models (TSFMs) under bi-level heterogeneity, addressing both inter-domain and intra-domain differences through adversarial local regularization, prototype alignment, and domain-aware aggregation. The problem setting is relevant, and reviewers generally acknowledge that the paper is technically sophisticated, clearly written, and supported by extensive experimental evaluation across point and probabilistic forecasting tasks, including large-scale and zero-shot settings.

The central issue is not empirical performance, but rather the clarity and necessity of the core motivation, as well as the alignment between the claimed contribution and the actual technical advance. Several reviewers expressed persistent concerns about whether heterogeneity should be framed as a primary obstacle in TSFM pretraining, and whether the observed gains truly stem from resolving heterogeneity rather than from implicit data balancing, regularization effects, or architectural choices.

While the rebuttal adds substantial empirical evidence (representation-level analysis, gradient conflict analysis, scaling studies, and leave-one-domain-out experiments), these additions primarily strengthen empirical support without fully resolving the conceptual disagreement. In particular, the notions of “domain-invariant representations,” “global dynamics,” and “domain awareness” remain somewhat abstract and loosely defined for time series data, where domain boundaries are often ambiguous and continuously varying. As a result, the motivation for enforcing domain invariance as a guiding principle remains debatable.

**Reviewer Concerns:**

The motivation for treating heterogeneity as a core problem in TSFM training is not universally convincing and remains conceptually contentious.

It is unclear whether performance gains arise from explicitly resolving heterogeneity or from secondary effects such as improved regularization or data balancing.

The framing as a general TSFM training framework does not fully align with the method’s nature as a specific representation learning and optimization strategy.

Despite strong experiments, the paper offers limited new conceptual insight into what constitutes “domain-invariant” or “globally shared” structure in time series.

**Reviewer Scores:**

Reviewer scores were mixed, spanning from marginally below to marginally above the acceptance threshold. During the discussion, one reviewer explicitly raised their score after the rebuttal, while others maintained borderline or negative assessments. Although the rebuttal improved confidence in the empirical soundness of the method, it did not lead to broad convergence toward acceptance.

---

### Decision · Program_Chairs · 2026-01-26

Reject